# TRANSFORMER PROTEIN LANGUAGE MODELS ARE UNSUPERVISED STRUCTURE LEARNERS

**Roshan Rao**[*]
UC Berkeley
rmrao@berkeley.edu

**Joshua Meier**
Facebook AI Research
jmeier@fb.com

**Tom Sercu**
Facebook AI Research
tsercu@fb.com

**Sergey Ovchinnikov**
Harvard University
so@g.harvard.edu

**Alexander Rives**
Facebook AI Research & New York University
arives@cs.nyu.edu

## ABSTRACT

Unsupervised contact prediction is central to uncovering physical, structural, and functional constraints for protein structure determination and design. For decades, the predominant approach has been to infer evolutionary constraints from a set of related sequences. In the past year, protein language models have emerged as a potential alternative, but performance has fallen short of state-of-the-art approaches in bioinformatics. In this paper we demonstrate that Transformer attention maps learn contacts from the unsupervised language modeling objective. We find the highest capacity models that have been trained to date already outperform a state-of-the-art unsupervised contact prediction pipeline, suggesting these pipelines can be replaced with a single forward pass of an end-to-end model.[1]

## 1 INTRODUCTION

Unsupervised modeling of protein contacts has an important role in computational protein design (Russ et al., 2020; Tian et al., 2018; Blazejewski et al., 2019) and is a central element of all current state-of-the-art structure prediction methods (Wang et al., 2017; Senior et al., 2020; Yang et al., 2019). The standard bioinformatics pipeline for unsupervised contact prediction includes multiple components with specialized tools and databases that have been developed and optimized over decades. In this work we propose replacing the current multi-stage pipeline with a single forward pass of a pre-trained end-to-end protein language model.

In the last year, protein language modeling with an unsupervised training objective has been investigated by multiple groups (Rives et al., 2019; Alley et al., 2019; Heinzinger et al., 2019; Rao et al., 2019; Madani et al., 2020). The longstanding practice in bioinformatics has been to fit linear models on focused sets of evolutionarily related and aligned sequences; by contrast, protein language modeling trains nonlinear deep neural networks on large databases of evolutionarily diverse and unaligned sequences. High capacity protein language models have been shown to learn underlying intrinsic properties of proteins such as structure and function from sequence data (Rives et al., 2019).

A line of work in this emerging field proposes the Transformer for protein language modeling (Rives et al., 2019; Rao et al., 2019). Originally developed in the NLP community to represent long range context, the main innovation of the Transformer model is its use of self-attention (Vaswani et al., 2017). Self-attention has particular relevance for the modeling of protein sequences. Unlike convolutional or recurrent models, the Transformer constructs a pairwise interaction map between all positions in the sequence. In principle this mechanism has an ideal form to model protein contacts.

In theory, end-to-end learning with a language model has advantages over the bioinformatics pipeline: (i) it replaces the expensive query, alignment, and training steps with a single forward

---

[*]Work performed during an internship at Facebook.

[1]Weights for all ESM-1 and ESM-1b models, as well as regressions trained on these models can be found at https://github.com/facebookresearch/esm.

pass, greatly accelerating feature extraction; and (ii) it shares parameters for all protein families, enabling generalization by capturing commonality across millions of evolutionarily diverse and unrelated sequences.

We demonstrate that Transformer protein language models learn contacts in the self-attention maps with state-of-the-art performance. We compare ESM-1b (Rives et al., 2020), a large-scale (650M parameters) Transformer model trained on UniRef50 (Suzek et al., 2007) to the Gremlin (Kamisetty et al., 2013) pipeline which implements a log linear model trained with pseudolikelihood (Balakrishnan et al., 2011; Ekeberg et al., 2013). Contacts can be extracted from the attention maps of the Transformer model by a sparse linear combination of attention heads identified by logistic regression. ESM-1b model contacts have higher precision than Gremlin contacts. When ESM and Gremlin are compared with access to the same set of sequences the precision gain from the protein language model is significant; the advantage holds on average even when Gremlin is given access to an optimized set of multiple sequence alignments incorporating metagenomics data.

We find a linear relationship between language modeling perplexity and contact precision. We also find evidence for the value of parameter sharing: the ESM-1b model significantly outperforms Gremlin on proteins with low-depth MSAs. Finally we explore the Transformer language model's ability to generate sequences and show that generated sequences preserve contact information.

## 2 BACKGROUND

**Multiple Sequence Alignments (MSAs)** A multiple sequence alignment consists of a set of evolutionarily related protein sequences. Since real protein sequences are likely to have insertions, deletions, and substitutions, the sequences are *aligned* by minimizing a Levenshtein distance-like metric over all the sequences. In practice heuristic alignment schemes are used. Tools like Jackhmmer and HHblits can increase the number and diversity of sequences returned by iteratively performing the search and alignment steps (Johnson et al., 2010; Remmert et al., 2012).

**Metrics** For a protein of length $L$, we evaluate the precision of the top $L$, $L/2$, and $L/5$ contacts for short range ($|i - j| \in [6, 12)$), medium range ($|i - j| \in [12, 24)$), and long range ($|i = j| \in [24, \infty)$) contacts. We also separately evaluate local contacts ($|i-j| \in [3, 6)$) for secondary structure prediction in Appendix A.9. In general, all contacts provide information about protein structure and important interactions, with shorter-range contacts being useful for secondary and local structure, while longer range contacts are useful for determining global structure (Taylor et al., 2014).

## 3 RELATED WORK

There is a long history of protein contact prediction (Adhikari & Cheng, 2016) both from MSAs, and more recently, with protein language models.

**Supervised contact prediction** Recently, supervised methods using deep learning have resulted in breakthrough results in *supervised* contact prediction (Wang et al., 2017; Jones & Kandathil, 2018; Yang et al., 2019; Senior et al., 2020; Adhikari & Elofsson, 2020). State-of-the art methods use deep residual networks trained with supervision from many protein structures. Inputs are typically covariance statistics (Jones & Kandathil, 2018; Adhikari & Elofsson, 2020), or inferred coevolutionary parameters (Wang et al., 2017; Liu et al., 2018; Senior et al., 2020; Yang et al., 2019). Other recent work with deep learning uses sequences or evolutionary features as inputs (AlQuraishi, 2018; Ingraham et al., 2019). Xu et al. (2020) demonstrates the incorporation of coevolutionary features is critical to performance of current state-of-the-art methods.

**Unsupervised contact prediction** In contrast to supervised methods, unsupervised contact prediction models are trained on sequences *without information from protein structures*. In principle this allows them to take advantage of large sequence databases that include information from many sequences where no structural knowledge is available. The main approach has been to learn evolutionary constraints among a set of similar sequences by fitting a Markov Random Field (Potts model) to the underlying MSA, a technique known as Direct Coupling Analysis (DCA). This was proposed by Lapedes et al. (1999) and reintroduced by Thomas et al. (2008) and Weigt et al. (2009).

Various methods have been developed to fit the underlying Markov Random Field, including mean-field DCA (mfDCA) (Morcos et al., 2011), sparse inverse covariance (PSICOV) (Jones et al., 2011) and pseudolikelihood maximization (Balakrishnan et al., 2011; Ekeberg et al., 2013; Seemayer et al., 2014). Pseudolikelihood maximization is generally considered state-of-the-art for unsupervised contact prediction and the Gremlin (Balakrishnan et al., 2011) implementation is used as the baseline throughout. We also provide mfDCA and PSICOV baselines. Recently deep learning methods have also been applied to fitting MSAs, and Riesselman et al. (2018) found evidence that factors learned by a VAE model may correlate with protein structure.

**Structure prediction from contacts**   While we do not perform structure prediction in this work, many methods have been proposed to extend contact prediction to structure prediction. For example, EVFold (Marks et al., 2011) and DCAFold (Sulkowska et al., 2012) predict co-evolving couplings using a Potts Model and then generate 3D conformations by directly folding an initial conformation with simulated annealing, using the predicted residue-residue contacts as constraints. Similarly, FragFold (Kosciolek & Jones, 2014) and Rosetta (Ovchinnikov et al., 2016) incorporate constraints from a Potts Model into a fragment assembly based pipeline. Senior et al. (2019), use features from a Potts model fit with pseudolikelihood maximization to predict pairwise distances with a deep residual network and optimize the final structure using Rosetta. All of these works build directly upon the unsupervised contact prediction pipeline.

**Contact prediction from protein language models**   Since the introduction of large scale language models for natural language processing (Vaswani et al., 2017; Devlin et al., 2019), there has been considerable interest in developing similar models for proteins (Alley et al., 2019; Rives et al., 2019; Heinzinger et al., 2019; Rao et al., 2019; Elnaggar et al., 2020; Lu et al., 2020; Madani et al., 2020; Shen et al., 2021). Rives et al. (2019) were the first to study protein Transformer language models, demonstrating that information about residue-residue contacts could be recovered from the learned representations by linear projections supervised with protein structures. Recently Vig et al. (2020) performed an extensive analysis of Transformer attention, identifying correspondences to biologically relevant features, and also found that different layers of the model are responsible for learning different features. In particular Vig et al. (2020) discovered a correlation between self-attention maps and contact patterns, suggesting they could be used for contact prediction.

Prior work benchmarking contact prediction with protein language models has focused on the supervised problem. Bepler & Berger (2019) were the first to fine-tune an LSTM pretrained on protein sequences to fit contacts. Rao et al. (2019) and Rives et al. (2020) perform benchmarking of multiple protein language models using a deep residual network fit with supervised learning on top of pretrained language modeling features.

In contrast to previous work on protein language models, we find that a state-of-the-art *unsupervised* contact predictor can be directly extracted from the Transformer self-attention maps. We perform a thorough analysis of the contact predictor, showing relationships between performance and MSA depth as well as language modeling perplexity. We also provide methods for improving performance using sequences from an MSA and for sampling sequences in a manner that preserves contacts.

## 4 MODELS

We compare Transformer models trained on large sequence databases to Potts Models trained on individual MSAs. While Transformers and Potts Models emerged in separate research communities, the two models share core similarities (Wang & Cho, 2019) which we exploit here. Our main result is that just as Gremlin directly represents contacts via its pairwise component (the weights), the Transformer also directly represents contacts via its pairwise component (the self-attention).

### 4.1 OBJECTIVES

For a set of training sequences, $X$, Gremlin optimizes the following pseudolikelihood loss, where a single position is masked and predicted from its context. Inputs are aligned, so all have length $L$:

$$\mathcal{L}_{\text{PLL}}(X; \theta) = \mathop{\mathbb{E}}_{x \sim X} \sum_{i=1}^{L} \log p(x_i | x_{j \neq i}; \theta) \tag{1}$$

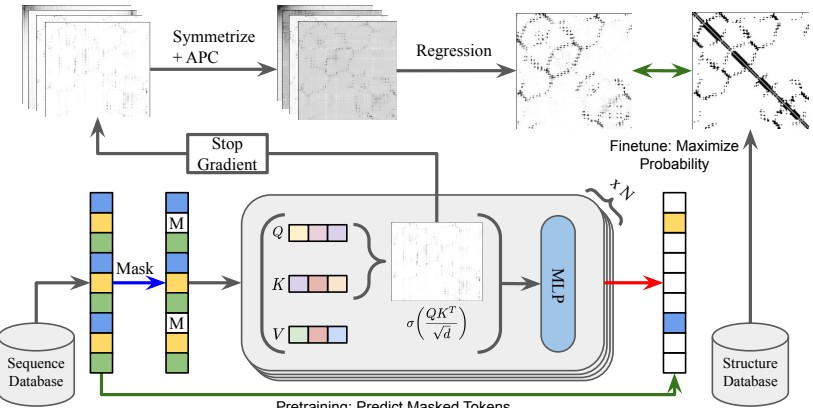

Figure 1: Contact prediction pipeline. The Transformer is first pretrained on sequences from a large database (Uniref50) via Masked Language Modeling. Once finished training, the attention maps are extracted, passed through symmetrization and average product correction, then into a regression. The regression is trained on a small number ($n \leq 20$) of proteins to determine which attention heads are informative. At test time, contact prediction from an input sequence can be done entirely on GPU in a single forward pass.

The masked language modeling (MLM) loss used by the Transformer models can be seen as a generalization of the Potts Model objective when written as follows:

$$\mathcal{L}_{\text{MLM}}(X;\theta) = \underset{x \sim X}{\mathbb{E}} \underset{\text{mask}}{\mathbb{E}} \sum_{i \in \text{mask}} \log p(x_i | x_{j \notin \text{mask}}; \theta) \tag{2}$$

In contrast to Gremlin, the MLM objective applied by protein language modeling is trained on unaligned sequences. The key distinction of MLM is to mask and predict multiple positions concurrently, instead of masking and predicting one at a time. This enables the model to scale beyond individual MSAs to massive sequence datasets. In practice, the expectation under the masking pattern is computed stochastically using a single sample at each epoch.

## 4.2 GREMLIN

The log probability optimized by Gremlin is described in section A.3. Contacts are extracted from the pairwise Gremlin parameters by taking the Frobenius norm along the amino acid dimensions, resulting in an $L \times L$ coupling matrix. Average product correction (APC) is applied to this coupling matrix to determine the final predictions (Appendix A.2).

Gremlin takes an MSA as input. The quality of the output predictions are highly dependent on the construction of the MSA. We compare to Gremlin under two conditions. In the first condition, we present Gremlin with all MSAs from the trRosetta training set (Yang et al., 2019). These MSAs were generated from all of Uniref100 and are also supplemented with metagenomic sequences when the depth from Uniref100 is too low. The trRosetta MSAs are a key ingredient in the state-of-the-art protein folding pipeline. See Yang et al. (2019) for a discussion on the significant impact of metagenomic sequences on the final result. In the second setting, we allow Gremlin access only to the same information as the ESM Transformers by generating MSAs via Jackhmmer on the ESM training set (a subset of Uniref50). See Appendix A.5 for Jackhmmer parameters.

## 4.3 TRANSFORMERS

We evaluate several pre-trained Transformer models, including ESM-1 (Rives et al., 2019), ProtBert-BFD (Elnaggar et al., 2020) and the TAPE Transformer (Rao et al., 2019). The key differences between these models are the datasets, model sizes, and hyperparameters (major architecture differences described in Table 3). Liu et al. (2019) previously showed that these changes can have a significant impact on final model performance. In addition to ESM-1, we also evaluate an updated version, ESM-1b, which is the result of a hyperparameter sweep. The differences are described in

Table 1: Average precision on 14842 test structures for Transformer models trained on 20 structures.

| Model | $6 \leq \text{sep} < 12$ | | | $12 \leq \text{sep} < 24$ | | | $24 \leq \text{sep}$ | | |
|---|---|---|---|---|---|---|---|---|---|
| | L | L/2 | L/5 | L | L/2 | L/5 | L | L/2 | L/5 |
| Gremlin (ESM Data) | 15.2 | 23.0 | 37.8 | 18.1 | 27.9 | 44.3 | 31.3 | 43.1 | 55.5 |
| mfDCA (trRosetta Data) | 16.3 | 23.7 | 35.8 | 19.7 | 29.8 | 45.5 | 33.0 | 43.5 | 54.2 |
| PSICOV[2] (trRosetta Data) | 15.4 | 23.6 | 39.2 | 18.3 | 28.4 | 45.7 | 32.6 | 45.2 | 58.1 |
| Gremlin (trRosetta Data) | 17.2 | 26.7 | 44.4 | 21.1 | 33.3 | 52.3 | 39.3 | 52.2 | 62.8 |
| TAPE | 9.9 | 12.3 | 16.4 | 10.0 | 12.6 | 16.6 | 11.2 | 14.0 | 17.9 |
| ProtBERT-BFD | 20.4 | 30.7 | 48.4 | 24.3 | 35.5 | 52.0 | 34.1 | 45.0 | 57.4 |
| ESM-1 (6 layer) | 11.0 | 13.2 | 15.9 | 11.5 | 14.6 | 19.0 | 13.2 | 16.7 | 21.5 |
| ESM-1 (12 layer) | 15.2 | 21.1 | 30.5 | 18.1 | 24.7 | 34.0 | 23.7 | 30.5 | 39.3 |
| ESM-1 (34 layer) | 20.3 | 30.2 | 46.0 | 23.8 | 34.3 | 49.2 | 34.7 | 44.6 | 56.0 |
| ESM-1b | **21.6** | **33.2** | **52.7** | **26.2** | **38.6** | **56.4** | **41.1** | **53.3** | **66.1** |

Section A.4. The Transformer processes inputs through a series of blocks alternating multi-head self-attention and feed-forward layers. In each head of a self-attention layer, the Transformer views the encoded representation as a set of query-key-value triples. The output of the head is the result of scaled dot-product attention:

$$\text{Attention}(Q, K, V) = \text{softmax}(QK^T/\sqrt{n}) \cdot V$$

Rather than only computing the attention once, the multi-head approach runs scaled dot-product attention multiple times in parallel and concatenates the output. Since self-attention explicitly constructs pairwise interactions ($QK^T$) between all positions in the sequence, the model can directly represent residue-residue interactions. In this work, we demonstrate that the $QK^T$ pairwise "self attention maps" indeed capture accurate contacts.

### 4.4 Logistic Regression

To extract contacts from a Transformer, we first pass the input sequence through the model to obtain the attention maps (one map for each head in each layer). We then symmetrize and apply APC to each attention map independently. The resulting maps are passed through an $L_1$-regularized logistic regression, which is applied independently at each amino acid pair $(i, j)$. At training time, we only train the weights of the logistic regression; we do not backpropagate through the entire model. At test time, the entire prediction pipeline can be run in a single forward pass, providing a single end-to-end pipeline for protein contact prediction that does not require any retrieval steps from a sequence database. See Appendix A.7 for a full description of the logistic regression setup.

## 5 Results

We evaluate models with the 15051 proteins in the trRosetta training dataset (Yang et al., 2019), removing 43 proteins with sequence length greater than 1024, since ESM-1b was trained with a context size of 1024. Of these sequences, Jackhmmer fails on 126 when we attempt to construct MSAs using the ESM training set (see Appendix A.5). This leaves us with 14882 total sequences. We reserve 20 sequences for training, 20 sequences for validation, and 14842 sequences for testing.

Table 1 shows evaluations of Gremlin, ESM-1, ESM-1b as well as the TAPE and ProtBERT-BFD models. Confidence intervals are within 0.5 percentage points for all statistics in Tables 1 and 2. In Table 1, all Transformer model contact predictors are trained with logistic regression on 20 proteins. We find that with only 20 training proteins ESM-1b has higher precision than Gremlin for short, medium, and long range contacts.

---

[2] PSICOV fails to converge on 24 sequences using default parameters. Following the suggestion in github.com/psipred/psicov, we increase $\rho$ to 0.005, 0.01, and thereafter by increments of 0.01, to a maximum of 0.1. PSICOV fails to converge altogether on 6 / 14842 sequences. We assign a score of 0 for these sequences.

Table 2: ESM-1b Ablations with limited supervision and with MSA information. $n$ is the number of logistic regression training proteins. $s$ is the number of sequences ensembled over.

| Model | Variant | $6 \leq \text{sep} < 12$ | | | $12 \leq \text{sep} < 24$ | | | $24 \leq \text{sep}$ | | |
|---|---|---|---|---|---|---|---|---|---|---|
| | | L | L/2 | L/5 | L | L/2 | L/5 | L | L/2 | L/5 |
| Gremlin | ESM Data | 15.2 | 23.0 | 37.8 | 18.1 | 27.9 | 44.3 | 31.3 | 43.1 | 55.5 |
| | trRosetta Data | 17.2 | 26.7 | 44.4 | 21.1 | 33.3 | 52.3 | 39.3 | 52.2 | 62.8 |
| ESM-1b (Ablations) | top-1 heads | 16.8 | 23.4 | 34.8 | 19.8 | 27.6 | 40.2 | 29.3 | 38.1 | 50.0 |
| | top-5 heads | 19.2 | 28.5 | 44.5 | 23.3 | 33.8 | 49.0 | 35.0 | 45.2 | 57.3 |
| | top-10 heads | 20.0 | 30.1 | 47.4 | 24.7 | 36.0 | 52.2 | 38.5 | 49.4 | 61.1 |
| | n=1, s=1 | 19.4 | 29.7 | 47.1 | 25.1 | 37.1 | 54.0 | 39.2 | 50.6 | 63.0 |
| | n=10, s=1 | 21.4 | 32.9 | 52.3 | 26.1 | 38.5 | 56.4 | 40.8 | 52.9 | 65.7 |
| | n=20, s=1 | 21.6 | 33.2 | 52.7 | 26.2 | 38.6 | 56.4 | 41.1 | 53.3 | 66.1 |
| | MSA, s=1 | 18.4 | 28.1 | 45.5 | 23.9 | 36.1 | 53.7 | 39.9 | 51.3 | 63.0 |
| ESM-1b ($s$ seqs) | n=20, s=16 | 21.9 | 33.8 | 53.6 | 26.7 | 39.4 | 57.5 | 41.9 | 54.3 | 67.3 |
| | n=20, s=32 | 22.0 | 34.1 | 54.0 | 26.9 | 39.8 | 58.1 | 42.3 | 54.8 | 67.8 |
| | n=20, s=64 | **22.1** | **34.3** | **54.3** | **27.1** | **40.1** | **58.5** | **42.6** | **55.1** | **68.2** |

In addition to this set, we also evaluate performance on 15 CASP13 FM Domains in Appendix A.6. On average ESM-1b has higher short, medium, and long range precision than Gremlin on all metrics, and in particular can significantly outperform on MSAs with low effective number of sequences. We also provide a comparison to the bilinear model proposed by Rives et al. (2020). The logistic regression model achieves a long-range contact precision at L of 18.6, while the fully supervised bilinear model achieves a long range precision at L of 20.1, an increase of only 1.5 points despite being trained on 700x more structures.

## 5.1 ABLATIONS: LIMITING SUPERVISION

While the language modeling objective is fully unsupervised, the logistic regression is trained with a small number of supervised examples. In this section, we study the dependence of the results on this supervision, providing evidence that the contacts are indeed learned in the unsupervised phase, and the logistic regression is only necessary to extract the contacts.

**Top Heads** Here we use the logistic regression only to determine the most important heads. Once they are selected, we discard the weights from the logistic regression and simply average the attention heads corresponding to the top-$k$ weight values. By taking the single best head from ESM-1b, we come close to Gremlin performance given the same data, and averaging the top-5 heads allows us to outperform Gremlin. Averaging the top-10 heads outperforms a full logistic regression on all other Transformer models and comes close to Gremlin given optimized MSAs.

**Low-N** The second variation we consider is to limit the number of supervised examples provided to the logistic regression. We find that with **only a single training example**, the model achieves a long range top-L precision of 39.2, which is statistically indistinguishable from Gremlin ($p > 0.05$). Using only 10 training examples, the model outperforms Gremlin on all the metrics. Since these results depend on the sampled training proteins, we also show a bootstrapped performance distribution using 100 different logistic regression models in Appendix A.10. We find that with 1 protein, performance can vary significantly, with long range top-L precision mean of 35.6, a median of 38.4, and standard deviation 8.9. This variation greatly decreases when training on 20 proteins, with a long range top-L precision mean of 40.1, median of 41.1, and standard deviation of 0.3. See Fig. 12 for the full distribution on all statistics.

**MSA Only** Finally, we consider supervising the logistic regression only with MSAs instead of real structures. This is the same training data used by the Gremlin baseline. To do this, we first train Gremlin on each MSA. We take the output couplings from Gremlin and mark the top $L$ couplings with sequence separation $\geq 6$ in each protein as true contacts, and everything else as false contacts,

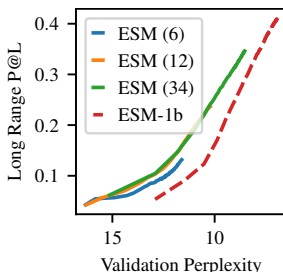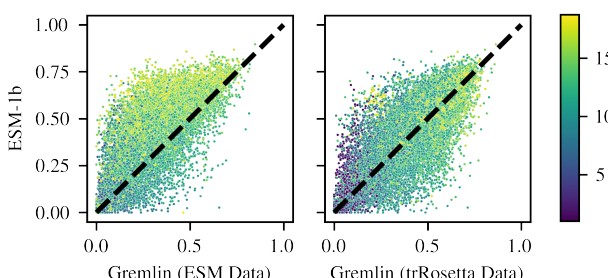

Figure 2: Left: Language modeling validation perplexity on holdout of Uniref50 vs. contact precision over the course of pre-training. ESM-1b was trained with different masking so perplexities between the versions are not comparable. Right: Long range P@L performance distribution of ESM-1b vs. Gremlin. Each point is colored by the log of the number of sequences in the MSA used to train Gremlin.

creating a binary decision problem. When trained on 20 MSAs, we find that this model achieves a long range P@L of 39.9, and generally achieves similar long range performance to Gremlin, while still having superior short and medium range contact precision.

## 5.2 ENSEMBLING OVER MSA

Transformer models are fundamentally single-sequence models, but we can further boost performance by ensembling predictions from multiple sequences in the alignment. To do so, we unalign each sequence in the alignment (removing any gaps), pass the resulting sequence through the Transformer and regression, and realign the resulting contact maps to the original aligned indices. For these experiments, we use the logistic regression weights trained on single-sequence inputs, rather than re-training the logistic regression on multi-sequence inputs. We also simply take the first $s$ sequences in the MSA. Table 2 shows performance improvements from averaging over 16, 32, and 64 sequences.

To better understand this result, we return to the single-sequence setting and study the change in prediction when switching between sequences in the alignment. We find that contact precision can vary significantly depending on the exact sequence input to the model, and that the initial query sequence of the MSA does not necessarily generate the highest contact precision (Fig. 9).

Lastly, Alley et al. (2019) presented a method of fine-tuning where a pretrained language model is further trained on the MSA of the sequence of interest ('evotuning'). Previously this has only been investigated for function prediction and for relatively low-capacity models. We fine-tune the full ESM-1b model (which has 50x more parameters than UniRep) on 380 protein sequence families. We find that after 30 epochs of fine-tuning, long range P@L increases only slightly, with an average of 1.6 percentage points (Fig. 16).

## 5.3 PERFORMANCE DISTRIBUTION

Although our model is, on average, better than Gremlin at detecting contacts, the performance distribution over all sequences in the dataset is still mixed. ESM-1b is consistently better at extracting short and medium range contacts (Fig. 7), but only slightly outperforms Gremlin on long range contacts when Gremlin has access to Uniref100 and metagenomic sequences. Fig. 2 shows the distribution of long range P@L for ESM-1b vs. Gremlin. Overall, ESM-1b has higher long range P@L on 55% of sequences in the test set.

In addition, we examine the relationship between MSA depth and precision for short, medium, and long range contacts (Fig. 3). Although our contact prediction pipeline does not make explicit use of MSAs, there is still some correlation between MSA depth and performance, since MSA depth is a measure of how many related sequences are present in the ESM-1b training set. We again see that ESM-1b consistently outperforms Gremlin at all MSA depths for short and medium range sequences. We also confirm that ESM-1b outperforms Gremlin for long range contact extraction for sequences with small MSAs (depth < 1000). ESM-1b also outperforms Gremlin on sequences with

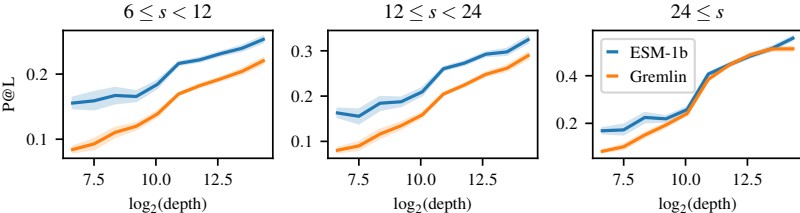

Figure 3: Gremlin (trRosetta) performance binned by MSA depth. For comparison, ESM-1b performance is also shown for the sequences in each bin.

the very largest MSAs (depth > 16000), which is consistent with prior work showing that Gremlin performance plateaus for very large MSAs and suggests that ESM-1b does not suffer from the same issues (Anishchenko et al., 2017).

## 5.4 LOGISTIC REGRESSION WEIGHTS

In Section 5.1 we show that selecting only a sparse subset of the attention heads can yield good results for contact prediction. Overall, the $L_1$-regularized logistic regression identifies 102 / 660 heads as being predictive of contacts (Fig. 6b). Additionally, we train separate logistic regressions to identify contacts at different ranges . These regressions identify an overlapping, but non-identical set of useful attention heads. Two attention heads have the top-10 highest weights for detecting contacts at all ranges. One attention head is highly positively correlated with local contacts, but highly negatively correlated with long range contacts. Lastly, we identify a total of 104 attention heads that are correlated (positively or negatively) with contacts at only one of the four ranges, suggesting that particular attention heads specialize in detecting certain types of contacts.

## 5.5 PERPLEXITY VS. CONTACT PRECISION

Fig. 2 explores the relationship between performance on the masked language modeling task (validataion perplexity) and contact prediction (Long Range P@L). A linear relationship exists between validation perplexity and contact precision for each model. Furthermore, for the same perplexity, the 12-layer ESM-1 model achieves the same long range P@L as the 34 layer ESM-1 model, suggesting that perplexity is a good proxy task for contact prediction. ESM-1 and ESM-1b models are trained with different masking patterns, so their perplexities cannot be directly compared, although a linear relationship is clearly visible in both. ESM-1 and ESM-1b have a similar number of parameters; the key difference is in their hyperparameters and architecture. The models shown have converged in pre-training, with minimal decrease in perplexity (or increase in contact precision) in the later epochs. This provides clear evidence that both model scale and hyperparameters play a significant role in a model's ability to learn contacts.

## 5.6 CALIBRATION, FALSE POSITIVES, AND ROBUSTNESS

One concern with neural networks is that, while they may be accurate on average, they can also produce spurious results with high confidence. We investigate this possibility from several perspectives. First, we find that logistic regression probabilities are close to true contact probability (mean-squared error = 0.014) and can be used directly as a measure of the model's confidence (Fig. 11).

Second, we analyze the false positives that the model does predict. We find that these are very likely to be within a Manhattan distance of 1-4 of a true contact (Fig. 13a). This suggests that false positives may arise due to the way a contact is defined (Cb-Cb distance within 8 angstroms), and could be marked as true contacts under a different definition (Zheng & Grigoryan, 2017). Further, when we explore an example where the model's predictions are not near a true contact, we see that the example in question is a homodimer, and that the model is picking up on inter-chain interactions (Fig. 14a). While these do not determine the structure of the monomer, they are important for its function (Anishchenko et al., 2017).

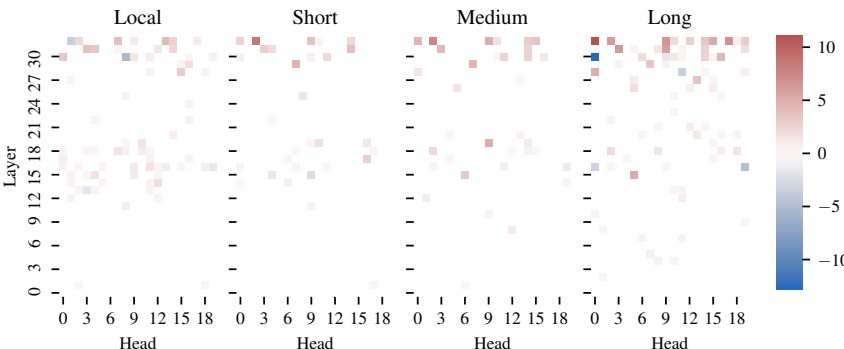

Figure 4: Logistic regression weights trained only on contacts in specific ranges: local [3, 6), short range [6, 12), medium range [12, 24), long range [24, ∞).

Third, we test the robustness of the model to insertions by inserting consecutive alanines at the beginning, middle, or end of 1000 randomly chosen sequences. We find that ESM-1b can tolerate up to 256 insertions at the beginning or end of the sequence and up to 64 insertions in the middle of the sequence before performance starts to significantly degrade. This suggests that ESM-1b learns a robust implicit alignment of the protein sequence. See Appendix A.12 for more details.

### 5.7 MSA Generation

Wang & Cho (2019) note that Transformers trained with the MLM objective can be used generatively. Here, we consider whether generations from ESM-1b preserve contact information. The ability to generate sequences that preserve this information is a necessary condition for generation of biologically active proteins (Hawkins-Hooker et al., 2021). We perform this evaluation by taking an input protein, masking out several positions, and re-predicting them. This process is repeated 10000 times to generate a pseudo-MSA for the input sequence (Algorithm 1). We feed the resulting MSA into Gremlin to predict contacts. Over all sequences from our test set, this procedure results in a long range contact P@L of 14.5. Fig. 17 shows one example where the procedure works well, with Gremlin on the pseudo-MSA having long range P@L of 52.2.

## 6 Discussion

Transformer protein language models trained with an unsupervised objective learn the tertiary structure of a protein sequence in their attention maps. Residue-residue contacts can be extracted from the attention by sparse logistic regression. Attention heads are found that specialize in different types of contacts. An ablation analysis confirms that the contacts are learned without supervision, and that the logistic regression is only necessary to extract the part of the model that represents contacts.

These results have implications for protein structure determination and design. The initial studies proposing Transformers for protein language modeling showed that representation learning could be used to derive state-of-the-art features across a variety of tasks, but were not able to show a benefit in the fully end-to-end setting (Rives et al., 2019; Rao et al., 2019; Elnaggar et al., 2020). For the first time, we show that protein language models can outperform state-of-the-art unsupervised structure learning methods that have been intensively researched and optimized over decades.

Finally, we establish a link between language modeling perplexity and unsupervised structure learning. A similar scaling law has been observed previously for supervised secondary structure prediction (Rives et al., 2019), and parallels observations in the NLP community (Kaplan et al., 2020; Brown et al., 2020). Evidence of scaling laws for protein language modeling support future promise as models and data continue to grow.

### Acknowledgments

We thank Justas Dauparas for valuable input and initial analysis. Sergey Ovchinnikov was supported by NIH Grant DP5OD026389.

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

Table 3: Major Architecture Differences in Protein Transformer Language Models

| Name | Layers | Hidden Size | Attn Heads | Parameters | Dataset |
|---|---|---|---|---|---|
| TAPE | 12 | 768 | 12 | 92M | Pfam |
| ProtBERT-BFD | 30 | 1024 | 16 | 420M | BFD100 |
| ESM-1 (6 layer) | 6 | 768 | 12 | 43M | Uniref50 |
| ESM-1 (12 layer) | 12 | 768 | 12 | 85M | Uniref50 |
| ESM-1 (34 layer) | 34 | 1280 | 20 | 670M | Uniref50 |
| ESM-1b | 33 | 1280 | 20 | 650M | Uniref50 |

# A  APPENDIX

## A.1  NOTATION

In the figures, we report contact precision in the range of $0.0$ to $1.0$. In the text and in the tables, we report contact precision in terms of percentages, in the range of 0 to 100.

## A.2  AVERAGE PRODUCT CORRECTION (APC)

In protein contact prediction, APC is commonly used to correct for background effects of entropy and phylogeny (Dunn et al., 2008). Given an $L \times L$ coupling matrix $F$, APC is defined as

$$F_{ij}^{\text{APC}} = F_{ij} - \frac{F_i F_j}{F} \tag{3}$$

Where $F_i$, $F_j$, and $F$ are the sum over the $i$-th row, $j$-th column, and the full matrix respectively. We apply APC independently to the symmetrized attention maps of each head in the Transformer. These corrected attention maps are passed in as input to a logistic regression.

## A.3  GREMLIN IMPLEMENTATION DETAILS

Gremlin is trained by optimizing the pseudolikelihood of $W$ and $V$, which correspond to pairwise and individual amino acid propensities. The pseudolikelihood approximation models the conditional distributions of the original joint distribution and can be written:

$$\log p(x_i^d = a | x_{j \neq i}^d; W_i, V_i) = \log \frac{\exp\left(V_{ia} + \sum_{j=1, j \neq i}^{N} \sum_{b=1}^{20} \mathbb{1}(x_j^d = b) W_{ijab}\right)}{\sum_{c=1}^{20} \exp\left(V_{ic} + \sum_{j=1, j \neq i}^{N} \sum_{b=1}^{20} \mathbb{1}(x_j^d = b) W_{ijcb}\right)} \tag{4}$$

subject to the constraint that $W_{ii} = 0$ for all $i$, and that $W_{ijab}$ is symmetric in both sequence $(i, j)$ and amino acid $(a, b)$. Additionally, Gremlin uses a regularization parameter that is adjusted based on the depth of the MSA.

## A.4  ESM-1 IMPLEMENTATION DETAILS

The original ESM-1 models were described in (Rives et al., 2019). ESM-1 is trained on Uniref50 in contrast to the TAPE model, which is trained on Pfam (Finn et al., 2014) and the ProtBERT-BFD model, which is trained on Uniref100 and BFD100 (Steinegger et al., 2019). ESM-1b is a new model, which is the result of an extensive hyperparameter sweep that was performed on smaller 12 layer models. ESM-1b is the result of scaling up that model to 33 layers.

Compared to ESM-1, the main changes in ESM-1b are: higher learning rate; dropout after word embedding; learned positional embeddings; final layer norm before the output; and tied input/output word embeddings. Weights for all ESM-1 and ESM-1b models can be found at `https://github.com/facebookresearch/esm`.

## A.5  JACKHMMER DETAILS

We use Jackhmmer version 3.3.1 with a bitscore threshold of 27 and 8 iterations to construct MSAs from the ESM training set. The failures on 126 sequences noted in Section 4.4 result from a segmentation fault in hmmbuild after several iterations (the number of successful iterations before the

Table 4: Average metrics on 15 CASP13 FM Targets. All baselines use MSAs generated via the trRosetta MSA generation approach.

| Model | Variant | $6 \leq \text{sep} < 12$ | | | $12 \leq \text{sep} < 24$ | | | $24 \leq \text{sep}$ | | |
|---|---|---|---|---|---|---|---|---|---|---|
| | | L | L/2 | L/5 | L | L/2 | L/5 | L | L/2 | L/5 |
| Baselines | mfDCA | 11.0 | 13.6 | 19.7 | 12.8 | 17.9 | 26.2 | 14.4 | 19.4 | 26.6 |
| | PSICOV[3] | 10.6 | 14.0 | 18.3 | 12.2 | 17.1 | 25.9 | 14.1 | 19.8 | 27.9 |
| | Gremlin | 12.1 | 16.1 | 23.6 | 14.5 | 20.8 | 32.5 | 16.8 | 23.4 | 28.5 |
| ESM-1b (attention) | top-1 heads | 11.8 | 15.8 | 23.8 | 17.0 | 20.8 | 29.6 | 13.6 | 17.9 | 22.7 |
| | top-5 heads | 15.3 | 20.9 | 29.6 | 18.7 | 27.0 | 33.0 | 14.6 | 20.6 | 26.8 |
| | top-10 heads | 16.6 | 22.7 | 32.1 | 21.8 | 29.5 | 39.8 | 17.9 | 23.2 | 30.4 |
| | n=1, s=1 | 16.4 | 23.5 | 34.7 | 23.0 | 30.8 | 41.6 | 18.1 | 23.3 | 29.9 |
| | n=10, s=1 | 18.6 | 25.3 | 39.3 | 24.1 | 31.9 | 41.4 | 18.7 | 25.2 | 33.2 |
| | n=20, s=1 | 19.3 | 26.6 | 37.0 | 24.0 | 31.5 | 40.2 | 18.6 | 25.0 | 33.8 |
| | MSA, s=1 | 14.2 | 20.3 | 30.5 | 21.0 | 29.1 | 42.3 | 18.4 | 23.7 | 31.5 |
| ESM-1b (bilinear) | n=20 | 14.1 | 17.7 | 19.8 | 17.7 | 20.9 | 27.9 | 11.2 | 13.9 | 17.0 |
| | n=14257 | **20.8** | **31.1** | **45.9** | **25.7** | **33.7** | **43.3** | **20.1** | **26.2** | **34.2** |

segmentation fault varies depending on the input sequence). Since we see this failure for less than 1% of the dataset we choose to ignore these sequences during evaluation.

Additionally, we evaluated alternate MSAs by running Jackhmmer until a Neff of 128 was achieved (with a maximum of 8 iterations), a procedure described by Zhang et al. (2020). This resulted in very similar, but slightly worse results (average long range P@L 29.3, versus 31.3 when always using the output of the eighth iteration). We therefore chose to report results using the 8 iteration maximum.

## A.6 RESULTS ON CASP13

In Table 4 we report results on the 15 CASP13 Free Modeling targets for which PDBs were publicly released. The specific domains evaluated are: T0950-D1, T0957s2-D1, T0960-D2, T0963-D2, T0968s1-D1, T0968s2-D1, T0969-D1, T0980s1-D1, T0986s2-D1, T0990-D1, T0990-D3, T1000-D2, T1021s3-D1, T1021s3-D2, T1022s1-D1. ESM-1b is able to outperform Gremlin, and simply averaging the top-10 heads of ESM-1b has comparable performance to Gremlin.

In addition, we compare our logistic regression model to the bilinear contact prediction model proposed by Rives et al. (2020). This model trains two separate linear projections of the final representation layer and computes contact probabilities via the outer product of the two projections plus a bias term, which generates the following unnormalized log probability:

$$\log p(\text{contact}) \propto (xW_1)(xW_2)^T + b \tag{5}$$

Here $x$ is a sequence-length vector of features in $\mathbb{R}^{L \times d}$. Each $W_i$ is a matrix in $\mathbb{R}^{d \times k}$, where $k$ is a hyperparameter controlling the projection size.

We train this model in both the limited supervision ($n = 20$) and full supervision ($n = 14257$) setting. For the limited supervision setting, we use the same 20 proteins used to train the sparse logistic regression model. For the full supervision setting we generate a 95/5% random training/validation split of the 15008 trRosetta proteins with sequence length $\leq 1024$.

We performed independent grid searches over learning rate, weight decay, and hidden size for the two settings. For the $n = 20$ setting, we found a learning rate of 0.001, weight decay of 10.0, and projection size of 512 had best performance on the validation set. For the $n = 14257$ setting we found a learning rate of 0.001, weight decay of 0.01, and projection size of 512 had best performance

---

[3]PSICOV fails to converge on 3 / 15 targets with default parameters. We follow the procedure suggested in `https://github.com/psipred/psicov` to increase `rho` to 0.005 for those domains.

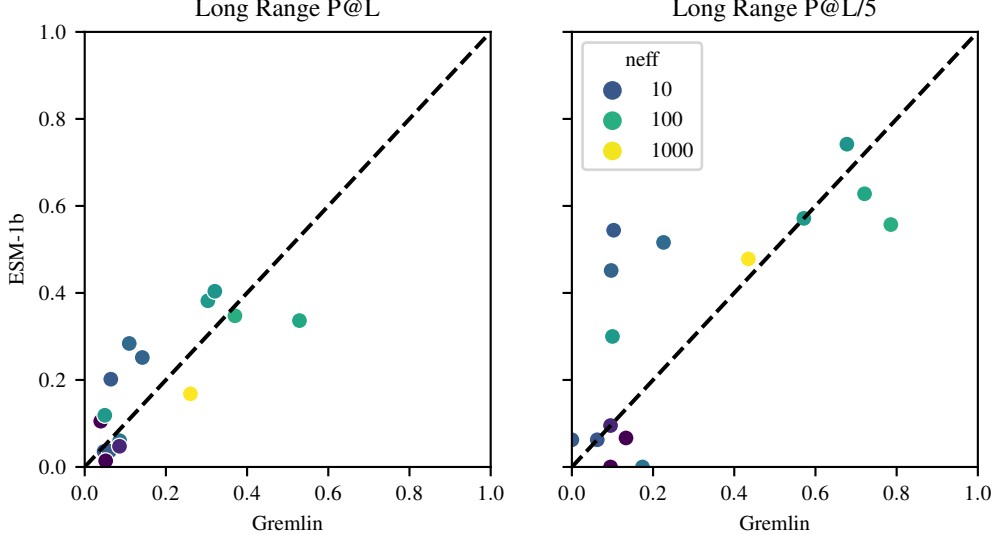

Figure 5: Results on 15 CASP13 FM Domains colored by Neff.

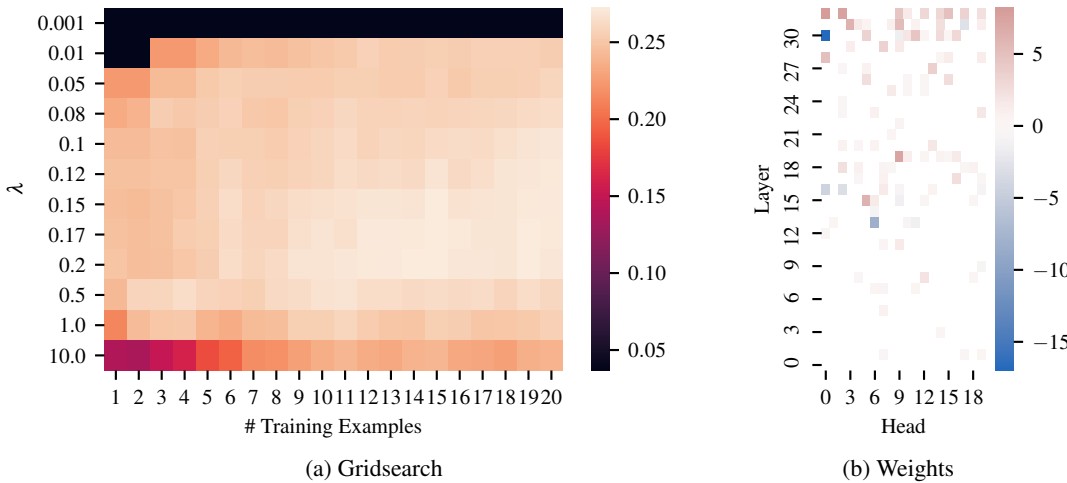

(a) Gridsearch

(b) Weights

Figure 6: (a) Gridsearch on logistic regression over number of training examples and number regularization penalty. Values shown are long range P@L over a validation set of 20 proteins. (b) Per-head and layer weights of the logistic regression on the best ESM-1b model.

on the validation set. All models were trained to convergence to maximize validation long range P@L with a patience of 10. The $n = 20$ models were trained with a batch size of 20 (i.e. 1 batch = 1 epoch) and the $n = 14257$ models were trained with a batch size of 128.

The bilinear model performs very poorly in the limited supervision setting, worse than simply taking the top-1 attention head. With full supervision, it moderately outperforms the logistic regression for an increase in long range P@L of 1.5 while using 700x more data.

In Fig. 5 we display results on the 15 FM targets colored by effective number of sequences. ESM-1b shows higher precision at L and L/5 on average, and is sometimes significantly higher for sequences with low Neff. Since ESM-1b training data was generated prior to CASP13, this suggests ESM-1b is able to generalize well to new sequences.

### A.7 Logistic Regression Details

Given a model with $M$ layers, $H$ heads, and an input sequence $x$ of length $L$, let $A_{mh}$ be the $L \times L$ contact map from the $h$-th head in the $m$-th layer. We first symmetrize this map and apply APC and let $a_{mhij}$ be the coupling weight between sequence position $i$ and $j$ in the resulting map. Then we define the probability of a contact between positions $i$ and $j$ according to a logistic regression with parameters $\beta$:

$$p(c_{ij}^d; \beta) = \frac{1}{1 + \exp\left(-\beta_0 - \sum_{m=1}^{M} \sum_{h=1}^{H} \beta_{mh} a_{mhij}^d\right)} \tag{6}$$

To fit $\beta$, let $\mathcal{D}$ be a set of training proteins, $k$ be a minimum sequence separation, and $\lambda$ be a regularization weight. The objective can then be defined as follows:

$$\mathcal{L}(\mathcal{D}; \beta) = \prod_{d \in \mathcal{D}} \prod_{i=1}^{L_d - k} \prod_{j=i+k}^{L_d} p(c_{ij}^d; \beta) \tag{7}$$

$$\hat{\beta} = \max_{\beta} \mathcal{L}(\mathcal{D}; \beta) + \frac{1}{\lambda} \sum_{m=1}^{M} \sum_{h=1}^{H} |\beta_{mh}| \tag{8}$$

We fit the parameters $\beta$ via scikit-learn (Pedregosa et al., 2011) and do not backpropagate the gradients through the attention weights. In total, our model learns $MH + 1$ parameters, many of which are zero thanks to the $L_1$ regularization.

There are three hyperparameters in our training setup: the number of proteins in our training set $\mathcal{D}$, the regularization parameter $\lambda$, and the minimum sequence separation of training contacts $k$. We find that performance improves significantly when increasing the $\mathcal{D}$ from 1 protein to 10 proteins, but that the performance gains drop off when $\mathcal{D}$ increases from 10 to 20 (Fig. 1). Through a hyperparameter sweep, we determined that the optimal $\lambda$ is 0.15. We find that ignoring local contacts ($|i - j| < 6$) is also helpful. Therefore, unless otherwise specified, all logistic regressions are trained with $|\mathcal{D}| = 20, \lambda = 0.15, k = 6$. See Fig. 6a for a gridsearch over the number of training proteins and regression penalty. We used 20 training proteins and 20 validation proteins for this gridsearch. Fig. 6b shows the weights of the final logistic regression used for ESM-1b.

### A.8 Performance Distribution

Fig. 7 shows the full distribution of performance of ESM-1b compared with Gremlin. When we provide Gremlin access to Uniref100, along with metagenomic sequences, ESM-1b still consistently outperforms Gremlin when extracting short and medium range contacts. For long range contacts, Gremlin is much more comparable, and has higher contact precision on 47% of sequences. With access to the same set of sequences, ESM-1b consistently outperforms Gremlin in detecting short, medium, and long range contacts. This suggests that ESM-1b can much better extract information from the same set of sequences and suggests that further scaling of training data may improve ESM-1b even further.

This analysis is further borne out in Fig. 8. Given the same set of sequences, ESM-1b outperforms Gremlin on average for short, medium, and long-range contacts regardless of the depth of the MSA generated from the ESM-1b training set.

Additionally, we find that ESM-1b can provide varying contact maps for different sequences in the MSA (Fig. 9). This is not possible for Gremlin, which is a family-level model. We leverage this in a fairly simple way to provide a modest boost to the contact precision of ESM-1b (Section 5.2).

### A.9 Secondary Structure

In Section 5.4 we show that some heads that detect local contacts (which often correspond to secondary structure) are actually negatively correlated with long range contacts. We test ESM-1b's ability to detect secondary structure via attention by training a separate logistic regression on the

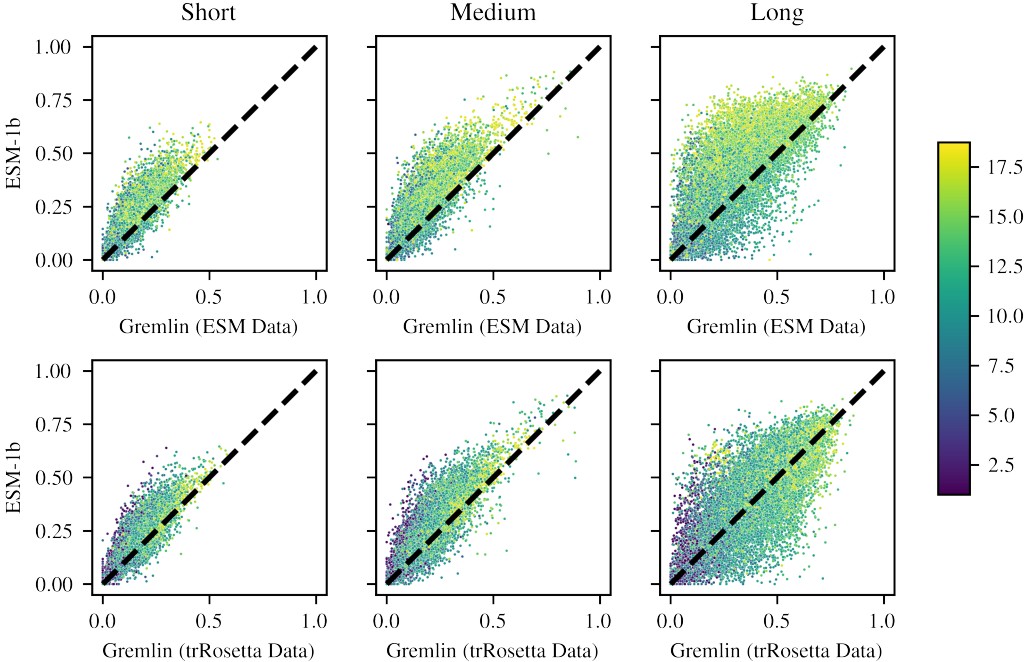

Figure 7: Short, medium, and long range P@L performance distribution of ESM-1b vs. Gremlin. Each point is colored by the $\log_2$ of the number of sequences in the MSA.

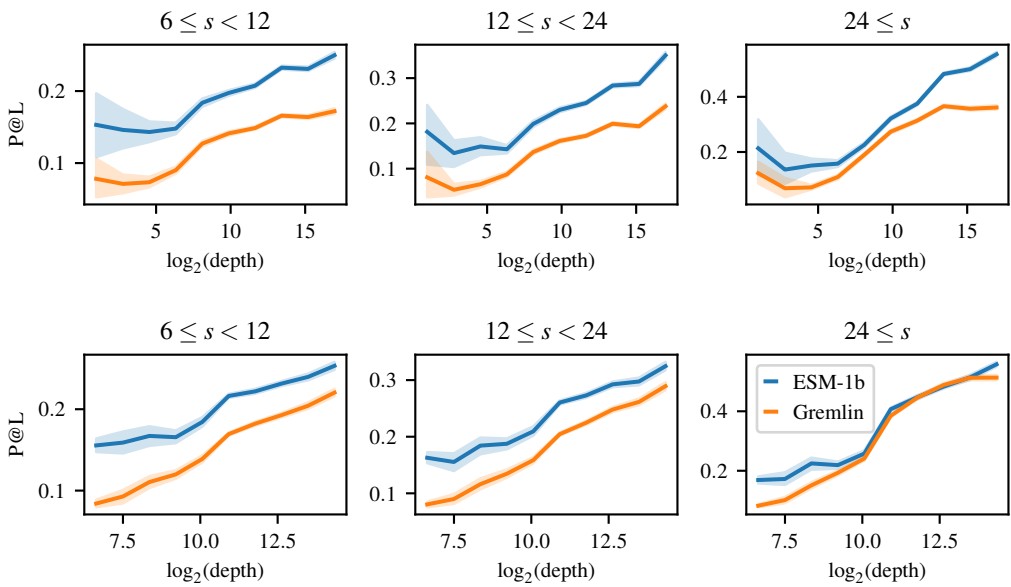

Figure 8: Gremlin performance binned by MSA depth using both ESM (top) and trRosetta (bottom) MSAs. For comparison, ESM-1b performance is also shown for the sequences in each bin.

Netsurf dataset (Klausen et al., 2019). As with the logistic regression on contacts, we compute attentions and perform APC + symmetrization. To predict the secondary structure of amino acid $i$, we feed as input the couplings $a_{mhij}$ for each layer $m$, for each head $h$, and for $j \in [i - 5, i + 5]$, for a total of 7260 input features. Using just 100 of the 8678 training proteins, we achieve 79.3% accuracy on 3-class secondary structure prediction on the CB513 test set (Cuff & Barton, 1999). Figure 10 shows the importance of each layer to predicting the three secondary structure classes.

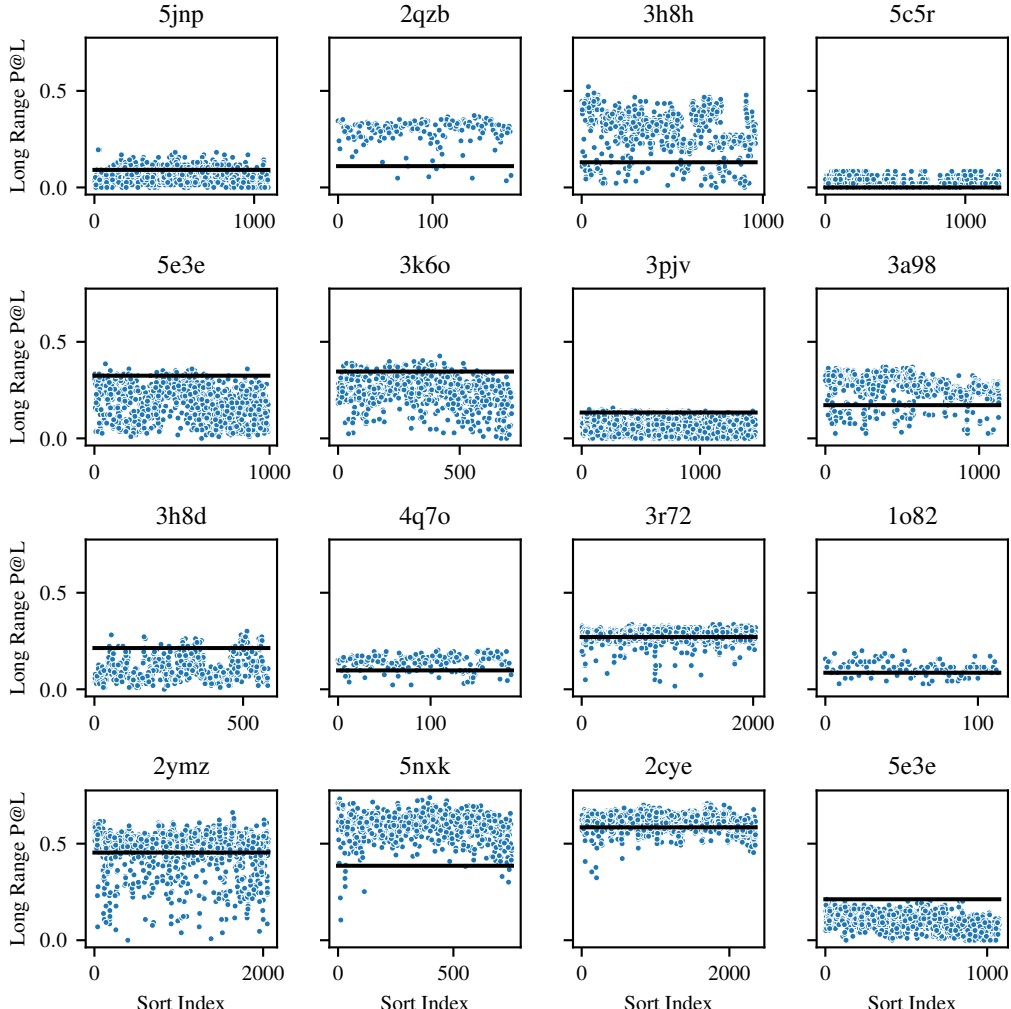

Figure 9: Distribution of contact perplexity when evaluating different sequences from the same MSA. The x-axis shows the index of each sequence, sorted in ascending order by hamming distance from the query sequence (query sequence is always index 0). The y-axis shows long range P@L. The black line indicates Gremlin performance on that MSA.

.

There are spikes in different layers for all three classes, indicating that particular heads within those layers are specializing in detecting specific classes of secondary structure.

Fig. 10 shows importance of each Transformer layer to predicting each of the three secondary structure classes. We see that, as with contact prediction, the most important layers are in the middle layers (14-20) and the final layers (29-33). Some layers spike more heavily on particular contact classes (e.g. layer 33 is important for all classes, but particularly important for $\beta$-strand prediction). This suggests that particular heads within these layers activate specifically for certain types of secondary structure.

## A.10 BOOTSTRAPPED LOW-N CONFIDENCE INTERVAL

Section 5.1 shows results from Low-N supervision on 1, 10, and 20 proteins. Since performance in this case depends on the particular proteins sampled we use bootstrapping to determine a confidence interval for each of these estimates. Using the full training, validation, and test set of 14882 proteins, we train 100 logistic regression models using a random sample of $N$ proteins, for $N = 1$, 10, and 20. Each model is then evaluated on the remaining $14882 - N$ proteins. The full distribution of samples

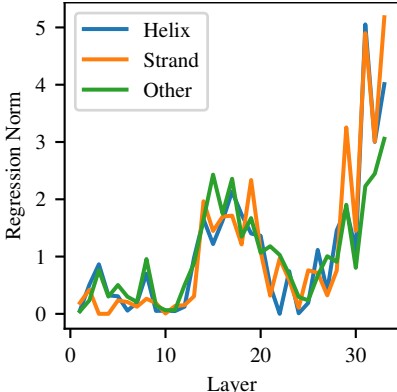

Figure 10: $L_2$ norm of weights for 3-class secondary structure prediction by Transformer layer.

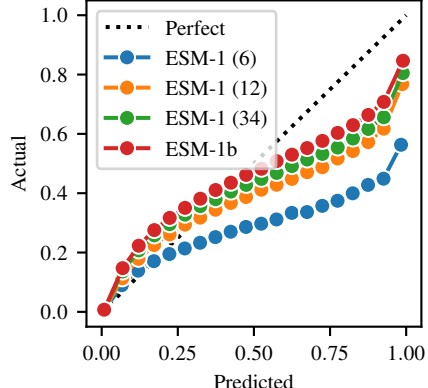

Figure 11: Calibrated probability of a real contact given predicted probability of contact over all test proteins.

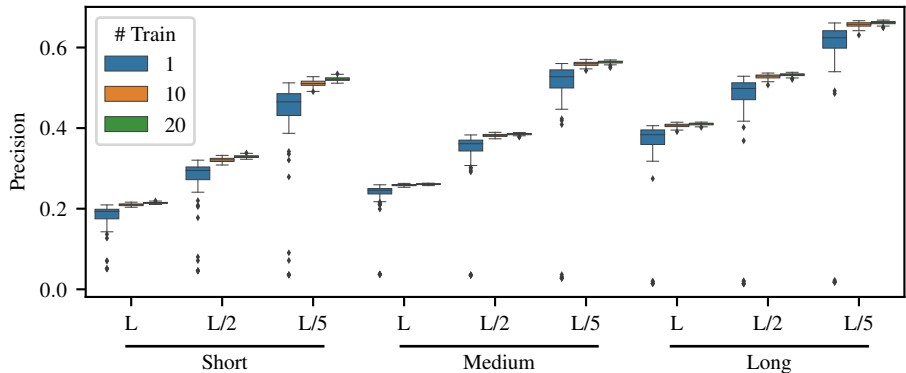

Figure 12: Distribution of precision for all reported statistics using 100 different logistic regression models. Each regression model is trained on a random sample of $N = 1, 10, 20$ proteins.

can be seen in Fig. 12. The confidence interval estimates for long range precision at L with 1, 10, and 20 training proteins are: $35.6 \pm 1.8$, $40.6 \pm 0.1$, and $41.0 \pm 0.1$ respectively.

## A.11  MODEL CALIBRATION AND FALSE POSITIVES

Vig et al. (2020) suggested that the attention probability from the TAPE Transformer was a well-calibrated estimator for the probability of a contact. In Fig. 11 we examine the same with the logistic regression trained on the ESM-1 and ESM-1b models. We note that ESM-1b, in addition to being more accurate overall than Gremlin, also provides actual probabilities.

We find that as with model accuracy, model calibration increases with larger scale and better hyperparameters. The 6, 12, and 34 layer ESM-1 models have mean-squared error of 0.074, 0.028, and 0.020 between predicted and actual contact probabilities, respectively. ESM-1b has a mean squared error of 0.014. Mean squared error is computed between contact probabilites split into 20 bins according to the scikit-learn calibration_curve function. It is therefore reasonable to use the logistic regression probability as a measure of the model's confidence.

In the case of false positive contacts we attempt to measure the Manhattan distance between the coordinates of predicted contacts and the nearest true contact (Fig. 13a). We observe that the Manhattan distance between the coordinates of false positive contacts are often very close (Manhattan distance between 1-4) to real contacts, and that very few false positives have a Manhattan distance $\geq 10$ from a true contact. With a threshold contact probability of 0.5, 83.8% of proteins have at least

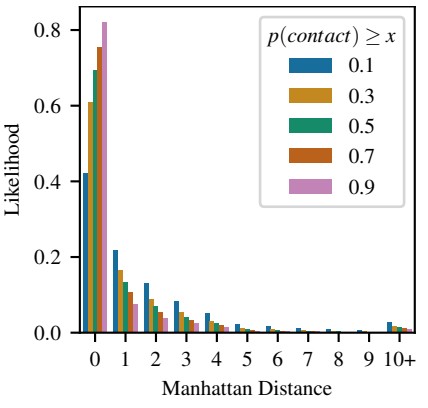 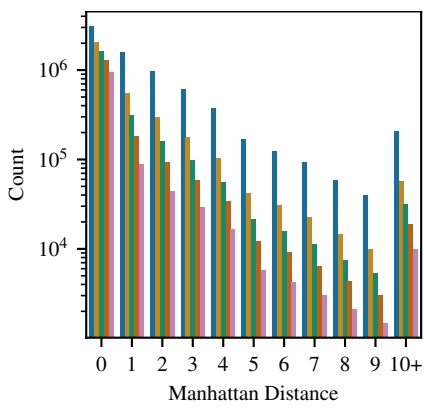

(a) Manhattan distance to nearest contact.          (b) Count of predictions by Manhattan distance.

Figure 13: (a) Distribution of Manhattan distance between the coordinates of predicted contacts and the nearest true contact at various thresholds of minimum $p(contact)$. A distance of zero corresponds to a true contact. (b) Actual counts of predictions by Manhattan distance across the full dataset (note y-axis is in log scale).

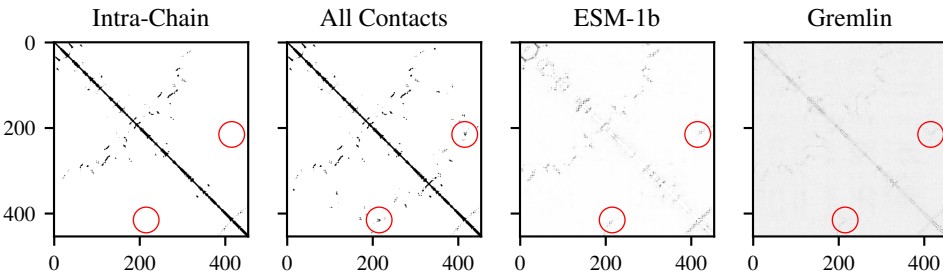

(a) Intra-chain, inter-chain, and predicted contacts for 5mlt, which is a homodimer.

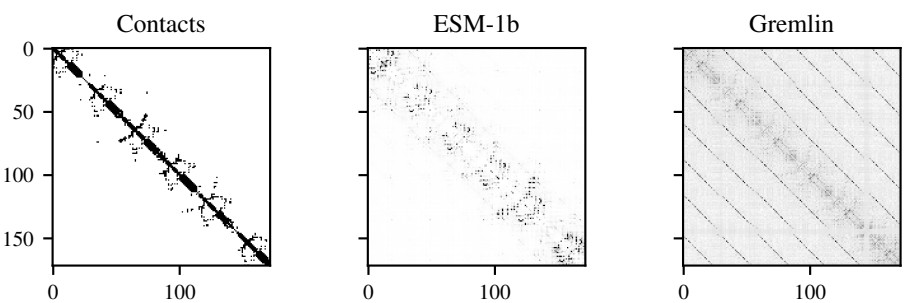

(b) Real and predicted contacts for the CTCF protein (pdbid: 5yel).

Figure 14: Illustration of two modes for ESM-1b where significant numbers of spurious contacts are predicted. (a) Predicted contacts which do occur in the full homodimer complex, but are not present as intra-chain contacts. (b) CTCF protein contacts. A small band of contacts near the 30-residue off-diagonal is predicted by ESM-1b. This band, along with additional similar bands are also predicted by Gremlin.

one predict contact with Manhattan distance > 4 to the nearest contact. This drops to 71.7% with a threshold probability of 0.7, and to 52.5% with a threshold probability of 0.9.

Fig. 14 highlights two modes for ESM-1b where signficant numbers of spurious contacts are predicted. Fig. 14a shows one example where the model does appear to hallucinate contacts around residues 215 and 415, which do not appear in the contact map for this protein. However, this protein

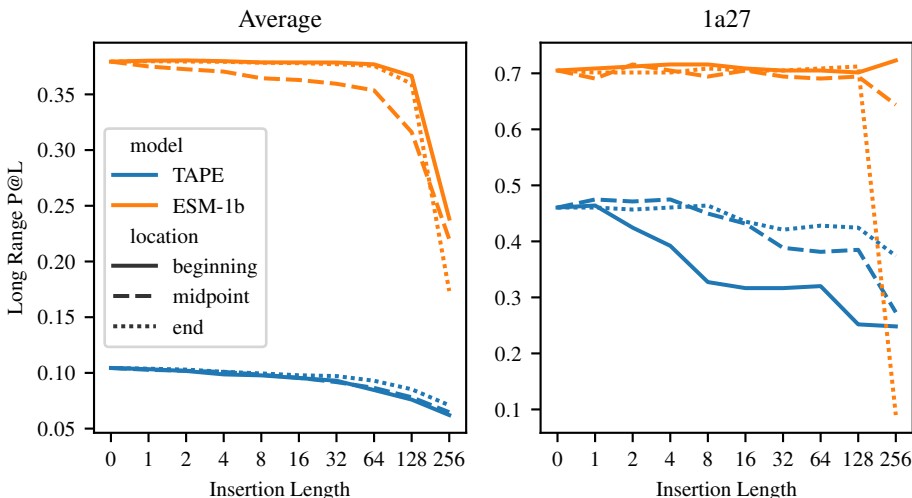

Figure 15: Robustness of ESM-1b and TAPE models to insertions of Alanine at the beginning, middle, and end of sequence

.

is a homodimer and these contacts are present in the inter-chain contact map. This suggests that some 'highly incorrect' false positives may instead be picking up on inter-chain contacts. Fig. 14b shows an example of a repeat protein, for which evolutionary coupling methods are known to pick up on additional 'bands' of contacts (Espada et al., 2015; Anishchenko et al., 2017). Multiple bands are visible in the Gremlin contact map, while only the first band, closest to the diagonal, is visible in the ESM-1b contact map. More analysis would be necessary to determine the frequency of these modes, along with additional potential modes.

## A.12 ALIGNMENT

One hypothesis as to the benefit of large language models as opposed to simpler Potts models is that they may be able to learn an implicit alignment due to their learned positional embedding. For a Potts Model, an alignment enables a model to relate positions in the sequence given evolutionary context despite the presence of insertions or deletions. We test the robustness of the model to insertions by inserting consecutive alanines at the beginning, middle, or end of 1000 randomly chosen sequences with initial sequence length < 512 (we limit initial sequence length in order to avoid out-of-memory issues after insertion). We find that ESM-1b can tolerate up to 256 insertions at the beginning or end of the sequence and up to 64 insertions in the middle of the sequence before performance starts to significantly degrade. This suggests that ESM-1b learns a robust implicit alignment of the protein sequence.

On the other hand, we find that the TAPE Transformer is less robust to insertions. On one sequence (pdbid: 1a27), we find the TAPE Transformer drops in precision by 12 percentage points after adding just 8 alanines to the beginning of the sequence, while ESM-1b sees minimal degradation until 256 alanines are inserted. We hypothesize that, because TAPE was trained on protein domains, it did not learn to deal with mis-alignments in the input sequence.

## A.13 EVOLUTIONARY FINETUNING DETAILS

We finetuned each model using a learning rate of 1e-4, 16k warmup updates, an inverse square root learning rate schedule, and a maximum of 30 epochs. This resulted in a varying number of total updates depending on the size of the MSA, with larger MSAs being allowed to train for more updates. This should ideally help prevent the model from overfitting too quickly on very small MSAs. We use a variable batch size based on the length of the input proteins, fixing a maximum of 16384 tokens per batch (so for a length 300 protein this would correspond to a batch size of 54). We use MSAs from trRosetta for finetuning all proteins with the exception of avGFP, where we use the same set of sequences from Alley et al. (2019).

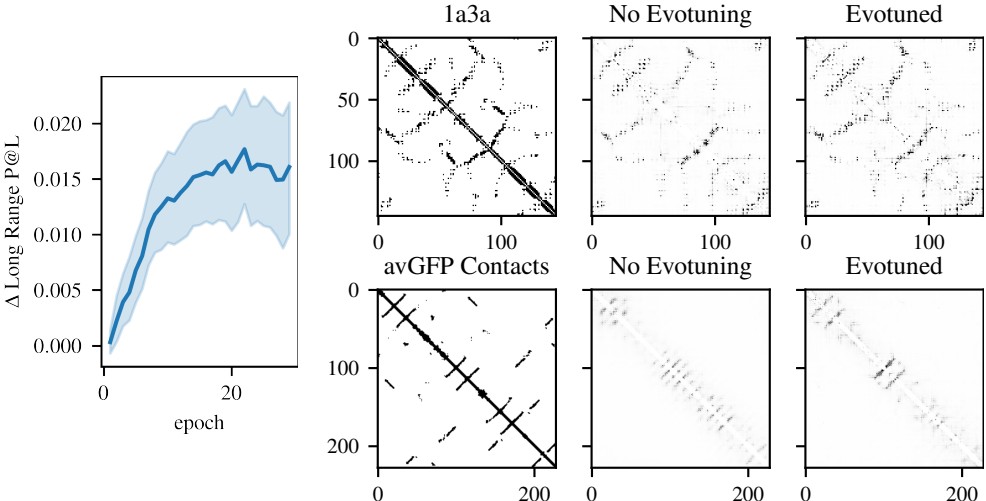

Figure 16: Left: Average change in contact precision vs. number of finetuning epochs over 380 proteins. Right: Real and predicted contacts before and after evolutionary finetuning for 1a3a and avGFP. For 1a3a, long range P@L improves from 54.5 to 61.4. For avGFP, long range P@L improves from 7.9 to 11.4.

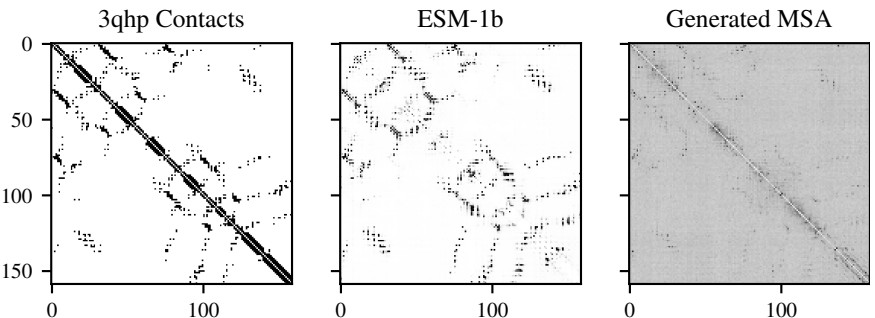

Figure 17: Contacts for 3qhp from Gremlin trained on pseudo-MSA generated by ESM-1b, compared to real and ESM-1b predicted contacts. The generated MSA achieves a long-range P@L of 52.2 while the attention maps achieve a precision of 76.7.

## A.14 MSA GENERATION

**Result:** Generated MSA
input // protein sequence
curr = input // optionally, the input can be repeated for batching
**for** $0 \leq i < 10000$ **do**
   masked = mask 20% of positions in curr;
   pred = model(masked);
   curr[masked positions] = pred[masked positions];
   MSA.append(curr);
   **if** $random() < 0.1$ **then**
     | curr = input;
   **end**
**end**

**Algorithm 1:** Quickly generate a pseudo-MSA from an input sequence.

Algorithm 1 presents the algorithm used to generate pseudo-MSAs from ESM-1b. Each pseudo-MSA is passed to GREMLIN in order to evaluate the preservation of contact information (Fig. 17).

