# OpenReview forum: "Transformer protein language models are unsupervised structure learners"
_ICLR.cc/2021/Conference — ICLR 2021 Poster_

### Official Review · AnonReviewer1 · 2020-10-17
**Using Transformers for protein contact prediction is not new**

**Rating:** 5
**Confidence:** 4

**Review:**

## Summary
The paper shows that Transformers trained unsupervised on millions of protein sequences learn information about protein contacts by using attention maps for contact prediction. The paper is mostly clearly written and discusses server interesting ablation experiments. However, two recent papers that appeared on arXiv before the ICLR submission deadlines also use Transformers for protein contact prediction. These papers and other methods for contact prediction beyond Gremlin are not described. I therefore consider the contributions as insufficient for an ICLR submission.

## Major comments

1. Using Transformer attention maps for protein contact prediction is not new. See Rives et al, 2020, ‘Biological structures and functions emerge…’, section 5.2, and Vig et al, 2020, ‘Bertology’ section 4.2. Both publications appeared on arXiv at least one month before the ICLR submission deadline and are not clearly discussed in the paper.

2. The introductions discusses existing work on Transformers for protein languages models. Existing methods for contact prediction (beyond Gremlin), however, are not described sufficiently.

3. It is unclear which sequences were used for training the Transformer models and how similar they are to test sequences.

4. The paper compares Transformers to Gremlin. However, it is unclear how well they perform to the CASP state-of-the art (see also Rives et al, 2020).

5. Section 3.4  does not describe clearly enough how attention maps were used for predicting contact maps. How were attention maps symmetrized? Which layers and heads were used and how were they aggregated? What is the number of resulting features that were used to train the logistic regression model? APC is not described or cited.

6. Section 4.5 discusses that Transformers can be also used for secondary structure prediction. This is not new (see Rives 2020 and Vig 2020) and does not fit well to the rest of the paper, which is about contact prediction.

6. Section 4.8: Using transformers for generating proteins with natural properties is not new (see Madani et al, 2020, ‘ProGen’ or Rives et al, 2020). ‘Wang & Cho’ were not the first who used Transformers generativity (see Vaswani, 2017).

---

> ### Author Response · Authors · 2020-11-12
> **Response to Reviewer 1**
>
> \>\>The paper shows that Transformers trained unsupervised on millions of protein sequences learn information about protein contacts by using attention maps for contact prediction...
>
> We thank the reviewer for their time, interest in the paper, and constructive feedback.
>
> \>\>However, two recent papers that appeared on arXiv before the ICLR submission deadlines also use Transformers for protein contact prediction. See Rives et al, 2020, ‘Biological structures and functions emerge…’, section 5.2, and Vig et al, 2020, ‘Bertology’ section 4.2.
>
> This is the first paper to show state-of-the-art results for unsupervised contact prediction from a transformer protein language model. Prior work Rives et al. 2020 benchmarks supervised contact prediction with deep residual networks. Vig et al. 2020 show that one specific head of the TAPE transformer is correlated with contacts (see Vig et al. 2020 Fig 4), but make no comparison to state-of-the-art methods for unsupervised contact prediction. In contrast, our work provides a new method that results in state-of-the-art performance on the unsupervised contact prediction problem.
>
> Moreover this is the first paper to demonstrate a state-of-the-art result for contact prediction from protein language modeling -- this is an important result for protein language modeling as previous work e.g. Rao et al. 2019, and Rives et al. 2020 have shown that protein language models fall well below state-of-the-art performance on supervised contact prediction tasks.
>
> \>\>These papers and other methods for contact prediction beyond Gremlin are not described...
>
> We will add a more thorough description of related work addressing unsupervised and supervised contact prediction.

---

> > ### Author Response · Authors · 2020-11-12
> > **Response to Reviewer 1 (Continued)**
> >
> > \>\>Major comments
> >
> > \>\>1. Using Transformer attention maps for protein contact prediction is not new...
> >
> > We address novelty of this work above and in the comment to all reviewers. Thank you for pointing out the need for a more extensive related work section, we agree and will add this section discussing both Rives, et al. 2020 and Vig, et al. 2020 as well as references suggested by the other reviewers.
> >
> > \>\>2. ...existing methods for contact prediction (beyond Gremlin), however, are not described sufficiently.
> >
> > We agree a more detailed background is necessary and will add this to the manuscript.
> >
> > \>\>3. It is unclear which sequences were used for training the Transformer models and how similar they are to test sequences.
> >
> > ESM transformers were trained using UniRef 50, which is noted in the introduction and in Figure 1. Note that the models and baselines are given access to the same set of sequences during test time. Since the number of similar sequences in the training set can be judged by the MSA depth of the sequence, we see in Fig 6 that as expected both the baseline and ESM perform better when there are more similar sequences available in the training set. However we show that ESM performs better than the baseline when fewer similar sequences are available in the ESM training set.
> >
> > We will also add new experiments on CASP13 proteins to the revision. Since the model was trained on data available prior to CASP13, these sequences will not be in the training set.
> >
> > \>\>4. ...it is unclear how well they perform to the CASP state-of-the art (see also Rives et al. 2020).
> >
> > We will add a comparison of unsupervised methods on CASP13 comparing the transformer attention maps, pseudolikelihood methods (Gremlin), mean field methods (Evcouplings), and sparse inverse covariance estimation (Psicov). We note that Gremlin is considered a state-of-the-art method for this problem.
> >
> > \>\>5. Section 3.4 does not describe clearly enough how attention maps were used for predicting contact maps...
> >
> > We provide a much more detailed explanation of the logistic regression in Appendix Section A.6. Attention maps are 2D matrices, so we symmetrize via 0.5 * (A + A^T). All layers and heads were used as input features, for a total of 660 features in the ESM-1b model (33 layers * 20 heads). We describe and cite APC in Appendix section A.2. Thank you for pointing out that APC is not cited in the main text -- we will add this citation.
> >
> > \>\>6. Section 4.5 discusses that Transformers can be also used for secondary structure prediction. This is not new...
> >
> > We agree that this is a peripheral result and for that reason the figures relating to secondary structure are in the appendix already. We thought it was interesting to describe as this is a different and more interpretable way to extract secondary structure from Transformer models than used in previous work e.g. Rives et al. 2019 and Vig et al. 2020 both of which did not use the attention maps. We note that local contacts (within a sequence separation of 6) can correspond to secondary structure, and so we use secondary structure as a proxy for analyzing the accuracy of contacts within this sequence separation range.
> >
> > \>\>7. Section 4.8: Using transformers for generating proteins with natural properties is not new (see Madani et al. 2020, ‘ProGen’ or Rives et al. 2020). ‘Wang & Cho’ were not the first who used Transformers generativity (see Vaswani, 2017).
> >
> > Vaswani et al. use an autoregressive decoder transformer as opposed to an encoder. We cite Wang & Cho as the first to generate from a non-autoregressive encoder transformer trained with a masked language modeling objective. Rives et al. 2019 and Rives et al. 2020 do not show results on generating proteins. Madani et al. 2020 show that it is possible to generate proteins that might preserve natural properties. However there are key differences. First, they use autoregressive decoder transformers, rather than bidirectional encoders. Our analysis demonstrates that bidirectional encoder transformers can also be used to generate proteins with natural properties. Additionally, our approach can generate proteins in the neighborhood of an existing protein, which may be highly useful for tasks such as protein engineering. Finally our analysis shows that generated sequences directly preserve the statistics needed to infer protein contacts which is not shown by Madani et al. 2020.

---

### Official Review · AnonReviewer3 · 2020-10-27
**Review for TRANSFORMER PROTEIN LANGUAGE MODELS ARE UNSUPERVISED STRUCTURE LEARNERS**

**Rating:** 7
**Confidence:** 5

**Review:**

In this paper, the authors show that transformer protein language models can learn protein contacts from the unsupervised language modelling objectives. They also show that the residue-residue contacts can be extracted by sparse logistic regression to learn coefficients on the attention heads. One of the advantages of using transformers models is that they do not require an alignment step nor the use of specialized bioinformatics tools (which are computationally expensive). When compared to a method based on multiple sequence alignment, the transformers models can obtain a similar or higher precision.

Contributions of this paper are:
- showing that the attention maps built in Transformer-based protein languages learn protein contacts, and when extracted, they perform competitively for protein contact prediction;
- a method for extracting attention maps from Transformer models;
- a comparison between a recent protein transformer protein language model (which does dot require sequence alignment), and a pseudo-likelihood-based optimization method that uses multiple sequence alignment;
- an analysis of how much the supervised learning (logistic regression) contributes to the results.

The paper covers a relevant topic and it is easy to read.

However, I have a number of concerns. The main contribution of the paper is that attention maps built in Transformer-based protein languages learn protein contacts and can be used for protein contact prediction. However, this was reported before in Rives et al.(2019) (doi: 10.1101/622803). Also, several methods have been developed for this problem, but are not included in the comparisons. Finally, the provided implementation details are not sufficient to reproduce the results of the paper.
I detail some of these concerns below, together with questions/suggestions for improvements:

1) I would recommend comparing transformers to other methods besides Gremlin, or justify why other methods were not included. This review can be helpful:

(Adhikari B, Cheng J., 2016.. doi: 10.1007/978-1-4939-3572-7_24)

Also, more recent methods that were published after the review are:

(Badri Adhikari, 2020. https://doi.org/10.1093/bioinformatics/btz593)

(Luttrell  et al., 2019. https://doi.org/10.1186/s12859-019-2627-6)

(Gao et al.,2019. https://doi.org/10.1038/s41598-019-40314-1)

(Ji S et al., 2019. https://doi.org/10.1371/journal.pone.0205214)

2) On page 7, the authors state that "We find that the logistic regression probabilities are reasonably well calibrated estimators of true contact probability and can be used directly as a measure of the model's confidence (Figure 10a)". However, from the plot in Figure 10a, it is not totally clear that the probabilities are well calibrated. Could the authors add more justifications of why they consider it well calibrated? Could they also show a comparison of the calibration of the other transformer models, perhaps using MSE as a calibration metric?

3) To understand the occurence of false positives, the authors analyze the Manhattan distance between the predicted contact and the true contact, which is between 1 and 4 for most false positives. They also show an example of a homodimer, for which predictions were far from the true contacts, and explain that the model is picking up inter-chain interactions. Could the authors report how many predictions have a Manhattan distance larger than 4? Is this one example representative of the group of false positives far from the true contact? Maybe the authors could analyse whether this happens in most of the cases.

4) While ESM-1 is open-source and publicly available, this is not the case for ESM-1b. In section A.5, the authors provide implementation details as differences between ESM-1 and ESM-1b, stating “Compared to ESM-1, the main changes in ESM-1b are: higher learning rate; dropout after word embedding; learned positional embeddings; final layer norm before the output; and tied input/output word embeddings. The weights of all ESM models throughout the training process were provided to us by the authors.”. In my opinion, this is not enough to reproduce the results in this paper. To make it reproducible, the authors need to provide a detailed enough description of the differences to make the reader able to implement ESM-1b, or provide the weights and hyperparameters required to reproduce their results.

---

> ### Author Response · Authors · 2020-11-12
> **Response to Reviewer 3**
>
> We thank the reviewer for their time, interest, and helpful critique.
>
> \>\>Contributions of this paper are [...] showing that the attention maps built in Transformer-based protein languages learn protein contacts, and when extracted, they perform competitively for protein contact prediction ...
>
> We agree on the contributions the reviewer has identified.
>
> \>\>However, I have a number of concerns.
>
> Below and in the comment to all reviewers we outline a plan to address these concerns.
>
> \>\>However, this was reported before in Rives et al (2019)
>
> Rives et al. 2019 does not report an analysis of attention maps. Rather, it uses the output from the final layer for supervised contact prediction.
>
> \>\>Also, several methods have been developed for this problem, but are not included in the comparisons.
>
> We believe that pseudolikelihood maximization (and by extension Gremlin, which implements this method) is the current state-of-the-art for unsupervised contact prediction. To address the concern we will add a comparison to the Evcouplings implementation of mean-field inference, and to the Psicov implementation of sparse inverse covariance matrix estimation.
>
> \>\>I would recommend comparing transformers to other methods besides Gremlin...
>
> Thank you for the suggestion. Section 2.2. of Adhikari 2016 describes evolutionary coupling-based methods. We note that Gremlin is an implementation of the pseudolikelihood based methods discussed in this section. The mean field approximation is also discussed here, as well as sparse inverse covariance matrix estimation. We propose adding comparisons to the mean field approximation (Evcouplings implementation) and the sparse inverse covariance matrix (Psicov implementation).
>
> \>\>Also, more recent methods that were published after the review are...
>
> These citations are for supervised contact prediction methods which are all deep neural networks trained with supervision from many protein structures. A comparison to our unsupervised contact prediction method is not appropriate as the problem settings are fundamentally distinct. We will add a discussion of supervised methods to the related work section.
>
> \>\>However, from the plot in Figure 10a, it is not totally clear that the probabilities are well calibrated...
>
> We agree that asserting the model is “well calibrated” is unclear without a baseline. Since it is not obvious what the correct baseline should be, we will reword this “we see that the model’s predicted probability is correlated with the actual contact probability.” We will add Pearson correlation between predicted and actual contact probability for the ESM1b model as well as the other transformer models as suggested.
>
> \>\>Could the authors report how many predictions have a Manhattan distance larger than 4...
>
> We appreciate the suggestions and will update the manuscript with the number of predictions with Manhattan distance greater than 4. After the submission deadline we have analyzed an alternate failure mode where the hallucinated contact is not representative of a true contact. We will add an example of this failure mode as well. We note that in both the old and new failure mode, hallucinated contacts appear in the Gremlin contacts as well. An analysis of which failure modes are most common is a very interesting idea, but would require too much manual work to be completed in the rebuttal period.
>
> \>\>To make it reproducible...
>
> We agree on the importance of reproducibility. We will make available contact prediction weights for ESM-1 and ESM-1b models allowing loading the models released by the ESM authors, along with a `predict_contacts` API. If you would like to review the code yourself, we will make the effort to anonymize it as much as possible.

---

> > ### Comment · AnonReviewer3 · 2020-11-24
> > **Substantial improvements to the manuscript**
> >
> > The authors have addressed all the comments appropriately and have made substantial modifications to the paper considering my comments.
> >
> > In my opinion, the related work section and the clear explanation on the supervised vs unsupervised contact prediction literature greatly improved the manuscript .
> >
> > I am still slightly concerned about reproducibility, as the modifications to the ESM architecture are not very clear to me. However, The authors have promised to share the weights, .
> >
> > In the new version of the paper it is still not clear how many predictions have a Manhattan distance larger than 4. It would be good if the authors could  provide a figure, or a table, detailing the distribution of predicted contacts and their respective Manhattan distances to the closest true contact. I appreciate that they provided the proportion of proteins with at least one predicted contact > 4 at different thresholds for contact probability, but in my opinion this does not give a clear enough picture.
> >
> > All things considered, I believe the paper has improved substantially, and I am willing to increase my score.

---

### Official Review · AnonReviewer2 · 2020-10-29
**Interesting analyses, but has overall limited novelty**

**Rating:** 6
**Confidence:** 4

**Review:**

**Summary**
The paper performs a number of analyses centered around the ability of transformer-based language models trained on protein sequence data to learn representations useful for predicting protein secondary and tertiary structure (the latter as contact maps). Specifically, the paper studies several pre-trained transformer models by fitting an L1-penalized logistic regression to amino acid pair contacts. Several experiments are performed to showcase that (i) transformer-based representations can outperform state-of-the art methods based on MSA in terms of contact prediction precision; (ii) that the necessary information for contact predictions in these representations is learned in an unsupervised manner (and not by the logistic regression put on top of these representations); and (iii) that the contact prediction probabilities are reasonably well calibrated.

**Score justification**
In its current form the paper presents interesting analyses, but has overall limited novelty. The ability of transformer models to learn representations predictive of secondary and tertiary structure has been demonstrated before (including in the papers proposing the models used by the authors). Furthermore, I have some questions regarding the methodology employed by the authors.


**Major comments**
* The main metric employed by the authors is the precision of the top L (protein length) contact prediction for a given range (P@L). I wonder why the authors do not also consider recall at L as an accompanying metric for reporting the results.
* When comparing ESM to the baseline Gremlin method, the authors consider two scenarios: (i) Gremlin trained on the trRosetta data; and (ii) Gremlin trained on the same data as the ESM transformer model. Overall, Gremlin trained on the ESM data - which is arguably the correct baseline for the ESM model -  performs worse than Gremlin trained on the trRosetta data. Why is that the case? How does the procedure for preparing MSA for the ESM data compare to that of the trRosetta data? Can it be tuned to improve Gremlin's performance?
* The paper compares several transformer models that differ primarily in the model size, dataset size and hyper-parameters. As can be seen from Table 1 of the manuscript, these differences are clearly important for the contact prediction task and thus should be summarized and discussed in more detail.
* From what I understand the sequences from the testing set of the contact prediction problem (or sequences highly similar to them) could appear in the training sets of the considered transformer models. This creates some information leakage. It's unclear from the results presented in the paper whether it is an issue or not - how does contact prediction precision / recall change as sequence similarity to the ESM training set drops?
* The authors present analysis on the usefulness of the representations learned by various attention heads for contact prediction; and on robustness of such predictions. I wonder how robust the results of these analyses are - they appear to have been performed using a single checkpoint of the ESM model, which is a result of stochastic training from random initialization.
* In the Appendix the authors talk about the benefit of using predicted contact maps for inferring the all-atom protein 3D structure. However no results on this are presented. I would be very eager to see the comparison of 3D structure accuracy inferred with ESM-predicted and Gremlin-predicted contacts.

**Minor comments**
* Introduction talks about the ESM-1b model but (as far as I can tell) a reference isn't provided until a later section.

---

> ### Author Response · Authors · 2020-11-12
> **Response to Reviewer 2**
>
> We thank the reviewer for their time and interest in the paper and for helpful comments.
>
> \>\>Several experiments are performed to showcase that (i) transformer-based representations can outperform state-of-the art methods based on MSA in terms of contact prediction precision; (ii) that the necessary information for contact predictions in these representations is learned in an unsupervised manner (and not by the logistic regression put on top of these representations); and (iii) that the contact prediction probabilities are reasonably well calibrated.
>
> \>\>In its current form the paper presents interesting analyses, but has overall limited novelty. The ability of transformer models to learn representations predictive of secondary and tertiary structure has been demonstrated before
>
> The reviewer’s main concern appears to be novelty. However we note that the reviewer agrees in point (i) above that the paper shows  “transformer-based representations can outperform state-of-the art methods based on MSA in terms of contact prediction precision.” No prior work has shown state-of-the-art performance for unsupervised contact prediction from a protein language model.
>
> Unsupervised contact prediction is an important and well studied problem (discussed in more depth in the comment to all reviewers) that has seen little progress since the introduction of pseudolikelihood maximization -- the state-of-the-art baseline we use in this paper. Additionally in point (ii) the reviewer agrees “that the necessary information for contact predictions in these representations is learned in an unsupervised manner.” This is also an important contribution of the work -- this paper is the first to show that state-of-the-art contacts are learned by Transformer language models in an unsupervised and interpretable manner.
>
> We realize we have not well situated the paper w.r.t. the supervised contact prediction literature, and prior work with protein language models in the supervised setting. We will endeavor to address this in the revision incorporating feedback from reviewers and additional references.

---

> > ### Author Response · Authors · 2020-11-12
> > **Response to Reviewer 2 (Continued)**
> >
> > Major comments
> >
> > \>\>The main metric
> > Precision at L is the standard contact prediction metric across the literature. Because having a small number of highly accurate contacts is useful (Skolnik et al. 1997, Kim et al. 2014), the field has standardized around this metric.
> >
> > \>\>When comparing ESM to the baseline Gremlin method
> >
> > Gremlin generally performs well when sequences are filtered using an identity cutoff of 80-90% similarity. As per Rives et al. 2020, ESM used a sequence identity clustering at 50% to train their model, which might not be optimal for Gremlin. The trRosetta data is our attempt to optimize Gremlin performance as much as possible. We believe this is a very strong baseline since these MSAs were used to achieve sota results for supervised contact prediction in Yang et al. 2020. Note it uses an optimal sequence identity cutoff for Gremlin, along with a series of e-value similarity thresholds. Finally, it augments smaller MSAs with additional metagenomic data.
> >
> > \>\>The paper compares several transformer models
> >
> > Thank you for the suggestion. We will add a table describing the differences in more detail and discussion to the paper describing how factors that vary between the models influence the results. We describe some ESM-1b changes in section A.5. The ESM-1b authors have made the model publicly available at https://github.com/facebookresearch/esm
> >
> > \>\>the sequences from the testing set of the contact prediction problem (or sequences highly similar to them) could appear in the training sets of the considered transformer models
> >
> > Please note that there isn’t an information leakage problem here in the sense that the baseline has access to the very same or strictly more sequences than our model was trained on. From Figure 7 we see that there is clearly a correlation between MSA depth and performance for both ESM and the baseline. MSA depth should provide a good proxy not just for sequence similarity (which is merely distance to the nearest sequence) but for the density of similar sequences present in the training set. We find that ESM outperforms the baseline significantly when the MSA depth is low (few similar sequences in the training dataset), and believe this is one of the strengths of our approach.
> >
> > To address this overlap more clearly, we will add results on CASP13 to the revision. Since training data for ESM-1b was generated prior to CASP13, these sequences will not have appeared in the training set.
> >
> > \>\>I wonder how robust the results of these analyses are
> >
> > In order to improve analysis of robustness, we will show bootstrapped results for training 100 different logistic regressions using randomly sampled training proteins.
> >
> > It is computationally expensive to train transformer protein language models, and so few are available for evaluation. We do show contact prediction results on multiple distinct transformer models, including models trained with different architectures by different groups on different data (ESM-1 6, 12, 34 layers, ESM-1b, ProtBERT-BFD, and TAPE).
> >
> > \>\>I would be very eager to see the comparison of 3D structure accuracy inferred with ESM-predicted and Gremlin-predicted contacts.
> >
> > We would also be very interested in comparing 3D structure inferred with ESM versus Gremlin contacts. This is likely beyond the scope of a revision and would be interesting for future work.

---

### Official Review · AnonReviewer4 · 2020-11-02
**Interesting idea, but background and comparisons are lacking**

**Rating:** 5
**Confidence:** 5

**Review:**

In this manuscript, the authors present a method for predicting residue-residue contacts within protein structures using the attention layers learned by transformer language models. Using the largest transformer language models trained to data, the authors show good performance for contact prediction. The paper is clearly written and easy to follow.

The general concept of fine tuning protein language models for contact prediction has circulated for some time which lessens the core contribution, but the authors approach is surprisingly data efficient and accurate. Overall this is an interesting work, though there is quite a bit of background on contact prediction missing. This paper is also very application specific and may not present new machine learning methods of general interest to the ICLR community. The existence of previous language model-based contact prediction methods reduces the novelty of this work, especially given that the model used here is from Rives et al. 2019, who already look at contact prediction. Furthermore, no comparisons with state-of-the-art evolutionary coupling-based or language model-based contact prediction methods are performed. With this in mind, the manuscript may be better suited to submission at a biology specific venue.

Additional specific comments follow below.

Major comments:
1.	Missing related work: there are a number of highly relevant prior works that are not mentioned/discussed. In particular, “Deep generative models of genetic variation capture the effects of mutations” – Riesselman et al. 2018 was, as far as I know, the first paper to show that deep generative models capture structure information (see Figure 6). Following that, “Learning protein sequence embeddings using information from structure” – Bepler & Berger 2019 was, to my knowledge, the first paper to propose deep language models (alignment free) for learning protein sequence representations and used those unsupervised representations for contact prediction. Furthermore, there has been extensive work in improving contact prediction using sequence + co-evolutionary features. See, for example, “Enhancing Evolutionary Couplings with Deep Convolutional Neural Networks” Liu et al. 2018 and “Accurate De Novo Prediction of Protein Contact Map by Ultra-Deep Learning Model” Wang et al. 2017. Other papers looking at protein structure prediction from sequence with deep learning, though they are less directly relevant, include “End-to-End Differentiable Learning of Protein Structure” AlQuraishi 2018 and “Learning Protein Structure with a Differentiable Simulator” Ingraham 2019.
2.	Before this work, others have looked at fine tuning language models for contact prediction. How do those approaches compare with the approach presented here? Rives et al look at contact prediction in their manuscript describing the transformer model (which is the same model used here) on CASP 11-13 (see Table 5 in their manuscript). How does that approach compare with this one? Likewise for Bepler & Berger
3.	Many methods have surpassed GREMLIN for contact prediction using evolutionary couplings. How do those approaches compare with this one? It would be helpful to see how this approach compares with truly state-of-the-art contact prediction methods. Reporting results on the CASP data would help to make this comparison.

Minor Comments:
1.	Although multiple sequence alignment methods have challenges especially as related to evolutionary coupling prediction, these methods have been heavily optimized for decades. The authors should provide citations for claimed failings such as “failure to find an optimal alignment” and “suboptimality of the substitution matrix and gap penalty.” Certainly, these may be sources of error in alignments, but I am not aware of any studies of the frequency or impacts of these errors on evolutionary coupling analysis. If these studies exist, I encourage the authors to cite them. If they do not exist, I suggest the authors focus on well known sources of error here (namely, alignment depth) and provide references.
2.	The authors use the language model without fine tuning, but the model could be fine tuned for each protein using its MSA. It’s great that contacts can be predicted without fine tuning, but it would be interesting to investigate whether additional gains can be made.
3.	Eight iterations of jackhmmer is a lot. In my personal experience, jackhmmer often diverges at 3+ iterations. By this I mean, the set of sequences and HMM learned by jackhmmer drift far away from the original sequence/family. Did the authors perform and quality checks of these alignments to ensure jackhmmer did not diverge?
4.	How are sequence depths in Figure 3 calculated? Is this the raw number of sequences in each MSA or is it after applying some sort of neighborhood weighting to calculate an effective number of sequences?

Things that would improve my rating:
1.	Provide a more comprehensive background review.
2.	Compare with state-of-the-art evolutionary coupling-based contact prediction methods.
3.	Compare with other language model-based contact prediction methods.
4.	What should interest the general machine learning community about this paper? What can we learn that might lead to better ML methods in the future? Convince me that this doesn’t belong in a bioinformatics venue!

---

> ### Author Response · Authors · 2020-11-12
> **Response to Reviewer 4**
>
> We thank the reviewer for their time and attention to the paper and for detailed comments.
>
> \>>The general concept of fine tuning protein language models for contact prediction has circulated for some time which lessens the core contribution,
> \>>The existence of previous language model-based contact prediction methods reduces the novelty of this work, especially given that the model used here is from Rives et al. 2019, who already look at contact prediction.
>
> While protein language models have been studied for contact prediction, e.g. Rives et al. 2019, Rao et al. 2019, this has been in the supervised setting. No existing work applies the models to the unsupervised contact prediction problem. This is the first work to demonstrate that unsupervised learning from a protein language model exceeds performance of state-of-the-art evolutionary couplings based unsupervised contact prediction.
>
> \>>Overall this is an interesting work, though there is quite a bit of background on contact prediction missing.
>
> Thank you for pointing out additional references. We will add a related work section covering contact prediction and other topics.
>
> \>>This paper is also very application specific and may not present new machine learning methods of general interest to the ICLR community.
> \>>With this in mind, the manuscript may be better suited to submission at a biology specific venue.
>
> We respectfully disagree. In this paper we propose an interpretable machine learning model that achieves state-of-the-art performance on an important unsupervised learning task in structural biology. This provides strong evidence that attention-based representations produced by unsupervised language modeling objectives can directly represent physical structures, which is of interest to the ICLR community.
>
> \>>Furthermore, no comparisons with state-of-the-art evolutionary coupling-based or language model-based contact prediction methods are performed.
>
> Pseudolikelihood maximization is the current state-of-the-art for unsupervised contact prediction (we use the Gremlin implementation). We will also add mean-field DCA (as implemented by Evcouplings) and sparse inverse covariance (Psicov implementation) as comparisons. There are no unsupervised language model-based contact prediction methods for comparison.

---

> > ### Author Response · Authors · 2020-11-12
> > **Response to Reviewer 4 (Continued)**
> >
> > Major comments
> > \>>1. Missing related work...
> >
> > Thank you for the suggestions and we agree that there is significant prior work in protein contact prediction that should be cited. We will add a Related Work incorporating these suggestions and additional references. While Risselman et al. 2018 and Bepler & Berger 2019 show that deep unsupervised models may learn structural information, our work demonstrates that this information is interpretable and accessible with little or no supervision required.
> >
> > \>>Before this work, others have looked at fine tuning language models for contact prediction...
> >
> > As far as we are aware, previous approaches using protein language models for contact prediction (including Bepler & Berger 2018, Rives et al. 2019, and Rao et al. 2019) consider the supervised contact prediction problem. Here, we focus on the model’s ability to learn contacts without supervision. In particular, our top-1, 5, and 10 head results show that the model does not require any supervision at all to predict contacts.
> >
> > \>>Many methods have surpassed GREMLIN for contact prediction using evolutionary couplings….
> >
> > Many supervised contact prediction methods have surpassed Gremlin for contact prediction, including those using evolutionary couplings, however we believe pseudolikelihood maximization (which Gremlin implements) is still considered state-of-the-art for unsupervised contact prediction. We will add results on the CASP data to make this point more clear.
> >
> > \>>Minor Comments:
> >
> > \>>Although multiple sequence alignment methods have challenges...
> >
> > Dickson & Gloor (2012) find that errors in the alignment can cause errors in downstream coevolution analyses. Malinverni & Barducci (2019) find that alignments that mix sub-families in an MSA cause errors in coevolution-based contact prediction. We will add references to these works and remove speculative comments.
> >
> > \>>1. The authors use the language model without fine tuning...on MSAs...
> >
> > We do perform this experiment (we call it “evolutionary finetuning” as proposed by Alley et al. 2019).  We discuss it briefly in Section 4.2 and in further detail in section A.11. We find that fine-tuning on individual MSAs leads to a minimal increase in performance, likely due to rapid overfitting of the large model. We note that this analysis is limited -- it is possible that fine tuning only certain layers or otherwise limiting the model’s ability to overfit may improve performance. We leave this to future work. We also show that averaging over sequences in the MSA can provide similar benefits without the costs of fine-tuning.
> >
> > \>>2. Eight iterations of jackhmmer is a lot...
> >
> > Thank you for this insightful comment. We propose to re-do this experiment following the procedure of Zhang et al. 2019, performing jackhmmer iterations until an Neff of 128 is reached.
> >
> > \>>3. How are sequence depths in Figure 3 calculated?
> >
> > We use the raw number of sequences in Figure 3.
> > Things that would improve my rating:
> >
> > \>>1. Provide a more comprehensive background review.
> >
> > Thank you for the references -- we agree and will include this in the revision.
> >
> > \>>2. Compare with state-of-the-art evolutionary coupling-based contact prediction methods.
> >
> > We believe that pseudolikelihood maximization as implemented by Gremlin is the current state-of-the-art unsupervised contact prediction method. We note that two new methods for fitting an MRF to an alignment have been proposed (Vorberg et al. 2018, Figliuzzi et al. 2018), but have been shown to have nearly identical performance to pseudolikelihood maximization.
> >
> > We specifically do not claim to achieve a state-of-the-art supervised contact prediction method. Instead, we claim that as with pseudolikelihood maximization, protein contacts naturally emerge from the unsupervised training signal in an interpretable and highly accessible manner. Therefore we do not believe that comparison to supervised methods, which incorporate significantly more information, is warranted.
> >
> > \>>3. Compare with other language model-based contact prediction methods.
> >
> > Same as response to 2. Since prior language model-based contact prediction methods are trained with thousands of structures, it would be inappropriate to compare this setting (large models with millions of parameters trained with thousands of protein structures) with the unsupervised setting (logistic regression fit with zero to twenty proteins); these are fundamentally different problem settings.
> >
> > \>>4. What should interest the general machine learning community about this paper?
> >
> > Please see the note above and in the response to all reviewers. We view this paper as arguing for a fundamentally different interpretation of learned representations in transformers, one that is highly interpretable and directly maps to physical structures.

---

### Author Response · Authors · 2020-11-12
**Response to All Reviewers**

We thank all reviewers for their thoughtful comments and suggestions. We are pleased to see that every reviewer considers this an interesting work.

In particular, we are glad that all reviewers appreciate that this work demonstrates transformers learn contacts in an unsupervised manner outperforming state-of-the-art unsupervised pipelines (R4: “surprisingly data efficient and accurate”, R2:  “contact predictions in these representations is learned in an unsupervised manner”; R3: “perform competitively for protein contact prediction [...] does not require sequence alignments”; R1: “learn information about protein contacts by using attention maps”).

The primary concern highlighted by all reviewers appears to be the novelty of this work. The reviewers point out that significant prior work exists around contact prediction from protein language models, e.g. Bepler and Berger 2019, Rives et al. 2019, Rao et al. 2019, Rives et al. 2020. We agree with the reviewers that protein language modeling has been applied to contact prediction in the past; however prior work focuses on the **supervised contact prediction** problem.

The main novelty of this work is that it focuses on the **unsupervised contact prediction** problem showing state-of-the-art performance. Our work is the first to propose an interpretable unsupervised contact prediction method from protein language models. The use of attention maps in our method distinguishes it from the approaches used in prior work in the supervised setting. Our approach is also completely different to all evolutionary-coupling methods for unsupervised contact prediction, is competitive with pseudolikelihood maximization at all MSA depths, and especially improves on pseudolikelihood maximization for shallow MSAs (see Fig 3).

Unsupervised contact prediction is well recognized as an important problem in its own right, evidenced by the breadth of prior work. Direct coupling analysis was initially described in Lapedes et al. 1999 and reintroduced by Thomas et al. 2008 and Weigt et al. 2009. Various methods have been developed to fit the underlying Markov Random Field, including inverse covariance (Morcos et al. 2011), sparse inverse covariance (Jones et al. 2012) and pseudolikelihood maximization (Balakrishnan et al. 2011, Seemayer et al. 2014, Ekeberg et al. 2013). Pseudolikelihood maximization is generally considered state-of-the-art for unsupervised contact prediction and is used as the baseline throughout.  In order to provide a more thorough comparison to prior methods, we will also add mean-field DCA and sparse inverse covariance as additional baselines.

No prior protein language modeling work directly considers the unsupervised contact prediction problem and benchmarks against the current state-of-the-art. Rives et al. 2019 fits linear projections and deep residual networks to the final hidden representation of the language model, demonstrating that information about contacts is encoded in the model and can be identified by supervision. Both Rao et al. 2019 and Rives et al. 2020 consider the supervised contact prediction application using deep residual networks and benchmark against supervised methods. Vig et al. 2020 Fig 4 shows that a particular head of the TAPE transformer correlates with contacts. They do not extract contact predictions using the attention maps, nor do they report contact precision based on attention maps. In Fig 16, they do report contact precisions, but these are fit from the hidden representations using supervision from many structures.

The reviewers have pointed out that a better discussion of prior work is needed. We acknowledge we have not properly positioned our work with respect to the literature on supervised contact prediction. We will add a thorough discussion incorporating the suggested references, discussing supervised and unsupervised approaches, and clearly delineating the unsupervised problem.

We believe this work is relevant to the ICLR community. Unsupervised learning is a core topic within the conference. The combination of unsupervised representation learning at scale with interpretability in a state-of-the-art method has broad interdisciplinary interest. It is of particular relevance to ICLR that representations learned from unlabeled sequence data map directly to underlying physical structures.

We propose the following plan to address the feedback from reviewers:
New related work section, with discussion of unsupervised and supervised contact prediction methods.
Additional unsupervised contact prediction baselines (mean field DCA, and sparse inverse covariance).
Results on CASP13 to enable easier comparison with other methods.
Misc additional experiments detailed inline in response to reviewer comments.

We appreciate the thoughtfulness of the reviewers and believe that these changes will significantly improve the paper. We welcome any additional comments or suggestions from the reviewers or broader community.

---

> ### Author Response · Authors · 2020-11-12
> **References**
>
> [1] Rives et al. (2020). Biological Structure and Function Emerge from Scaling Unsupervised Learning to 250 Million Protein Sequences.
>
> [2] Rao et al. (2019). Evaluating Protein Transfer Learning with TAPE.
>
> [3] Vig et al. (2020). BERTology Meets Biology: Interpreting Attention in Protein Language Models.
>
> [4] Bepler & Berger (2019). Learning protein sequence embeddings using information from structure.
>
> [5] Hopf et al. (2018). The EVcouplings Python framework for coevolutionary sequence analysis.
>
> [6] Lapedes et al. (1999). Correlated Mutations in Models of Protein Sequences: Phylogenetic and Structural Effects.
>
> [7] Thomas et al. (2008). Graphical Models of Residue Coupling in Protein Families.
>
> [8] Weigt et al. (2009). Identification of direct residue contacts in protein-protein interaction message passing.
>
> [9] Jones et al. (2012). PSICOV: precise structural contact prediction using sparse inverse covariance estimation on large multiple sequence alignments.
>
> [10] Alley et al. (2019). Unified rational protein engineering with sequence-based deep representational learning.
>
> [11] Zhang et al. (2019). DeepMSA: constructing deep multiple sequence alignment to improve contact prediction and fold-recognition for distant-homology proteins.
>
> [12] Skolnik et al. (1997). MONSSTER: a method for folding globular proteins with a small number of distance restraints.
>
> [13] Kim et al. (2014). One contact for every twelve residues allows robust and accurate topology-level protein structure modeling.
>
> [14] Dickson & Gloor (2012). Protein Sequence Alignment Analysis by Local Covariation: Coevolution Statistics Detect Benchmark Alignment Errors.
>
> [15] Malinverni & Barducci (2019). Coevolutionary Analysis of Protein Subfamilies by Sequence Reweighting.
>
> [16] Yang et al. (2020). Improved protein structure prediction using predicted interresidue orientations.
>
> [17] Vaswani et al. (2017). Attention is all you need.
>
> [18] Wang & Cho (2018). BERT has a Mouth, and It Must Speak: BERT as a Markov Random Field Language Model.
>
> [19] Madani et al. (2020). ProGen: Language Modeling for Protein Generation.
>
> [20] Balakrishnan et al. (2011). Learning generative models for protein fold families.
>
> [21] Seemayer et al. (2014). CCMpred--fast and precise prediction of protein residue-residue contacts from correlated mutations.
>
> [22] Vorberg et al. (2018). Synthetic protein alignments by CCMgen quantify noise in residue-residue contact prediction.
>
> [23] Figliuzzi et al. (2018). How pairwise coevolutionary models capture the
> collective residue variability in proteins.
>
> [24] Morcos et al. (2011). Direct-coupling analysis of residue coevolution captures native contacts across many protein families.
>
> [25] Ekeberg et al. (2013). Improved contact prediction in proteins: Using pseudolikelihoods to infer Potts models.

---

### Author Response · Authors · 2020-11-15
**Updated Revisions**

In order to address concerns regarding related work and evaluation, we have updated our submission with the following changes:

1. New Related Work section, describing unsupervised and supervised contact prediction along with relevant citations.
2. CASP 13 Evaluations (Section A.6, Table 4, and Figure 5).
3. Comparisons with mfDCA and PSICOV baselines.
4. Comparisons with Rives et al. 2020 supervised bilinear model on CASP13.
5. Computed mean squared error for calibration analysis, and show comparison on ESM models.
6. Removed speculative wording around sources of alignment failures.
7. Bootstrapping analysis (Section A.10)
8. Secondary structure analysis moved to appendix
9. Additional discovered mode of false-positive contacts shown

---

### Decision · Program_Chairs · 2021-01-07
**Final Decision**

**Decision:**

Accept (Poster)

**Comment:**

The authors have done a very thorough job of responding to the comments from reviewers. The paper has a clear contribution, namely that attention maps predict contacts as well as existing unsupervised pipelines. This paper deserves to be published.

In the final version, the authors should discuss briefly "BERTology Meets Biology: Interpreting Attention in Protein Language Models"(https://openreview.net/forum?id=YWtLZvLmud7) and "Improving Generalizability of Protein Sequence Models via Data Augmentations" (https://openreview.net/forum?id=Kkw3shxszSd). However, the authors should also make sure that the final version respects the ICLR length limits.

I am recommending poster acceptance because the results are anticlimactic given the recent success of Deepmind at CASP 2020.